# Munc18-1-regulated stage-wise SNARE assembly underlying synaptic exocytosis

Lu Ma[1], Aleksander A Rebane[1,2,3], Guangcan Yang[1,4], Zhiqun Xi[1], Yuhao Kang[1], Ying Gao[1], Yongli Zhang[1]*

[1]Department of Cell Biology, Yale School of Medicine, New Haven, United States; [2]Integrated Graduate Program in Physical and Engineering Biology, Yale University, New Haven, United States; [3]Department of Physics, Yale University, New Haven, United States; [4]Department of Physics, Wenzhou University, Wenzhou, China

**Abstract** Synaptic-soluble *N*-ethylmaleimide-sensitive factor *a*ttachment *r*eceptor (SNARE) proteins couple their stage-wise folding/assembly to rapid exocytosis of neurotransmitters in a Munc18-1-dependent manner. The functions of the different assembly stages in exocytosis and the role of Munc18-1 in SNARE assembly are not well understood. Using optical tweezers, we observed four distinct stages of assembly in SNARE N-terminal, middle, C-terminal, and linker domains (or NTD, MD, CTD, and LD, respectively). We found that SNARE layer mutations differentially affect SNARE assembly. Comparison of their effects on SNARE assembly and on exocytosis reveals that NTD and CTD are responsible for vesicle docking and fusion, respectively, whereas MD regulates SNARE assembly and fusion. Munc18-1 initiates SNARE assembly and structures t-SNARE C-terminus independent of syntaxin N-terminal regulatory domain (NRD) and stabilizes the half-zippered SNARE complex dependent upon the NRD. Our observations demonstrate distinct functions of SNARE domains whose assembly is intimately chaperoned by Munc18-1.

*For correspondence: yongli.zhang@yale.edu

**Competing interests:** The authors declare that no competing interests exist.

## Introduction

SNAREs are evolutionarily conserved molecular machines that drive fusion of transport vesicles with their target membranes, thereby transferring materials and information between different cells or cellular compartments (*Sudhof and Rothman, 2009*; *Wickner and Schekman, 2008*; *Jahn and Fasshauer, 2012*). In the cell, SNARE-mediated membrane fusion requires cognate cytoplasmic Sec1p/ Munc18 (SM) proteins (*Verhage et al., 2000*). Mutations in SNARE and SM proteins have been associated with many important human diseases (*Sudhof, 2014*; *Shen et al., 2014*; *Saitsu et al., 2008*; *Shen et al., 2015*).

Fusion of synaptic vesicles with the pre-synaptic plasma membrane mediates neurotransmitter release at neuronal and neuromuscular junctions (*Sudhof and Rothman, 2009*). The fusion requires the vesicle-anchored v-SNARE VAMP2 (also called synaptobrevin), the plasma-membrane-associated t-SNAREs syntaxin-1 and SNAP-25 (*Figure 1A*), and the SM-protein Munc18-1 (*Sollner et al., 1993*; *Verhage et al., 2000*; *Sudhof, 2014*; *Rothman, 2014*). Extensive evidence suggests that the t- and v-SNAREs first form a partial trans-SNARE complex bridging the two membranes with the assistance of many regulatory proteins, including Munc18-1, synaptotagmin, complexin, and Munc13, which docks the vesicle to the plasma membrane (*Xu et al., 1999*; *Melia et al., 2002*; *Gao et al., 2012*; *Sudhof, 2014*; *Rothman, 2014*). Upon the arrival of an action potential and resultant $Ca^{2+}$ influx, the trans-SNARE complex rapidly zippers to complete its assembly and membrane fusion. However, an alternative model suggests that SNARE assembly only starts after $Ca^{2+}$ triggering and completes in one step without any intermediates (*Jahn and Fasshauer, 2012*). The fully assembled SNARE complex forms a stable parallel four-helix bundle (*Sutton et al., 1998*; *Stein et al., 2009*)

**eLife digest** Plants, animals and other eukaryotes transport many large molecules within their cells inside membrane-bound packages called vesicles. These vesicles can fuse with the membrane of a target compartment in the cell to deliver their contents inside, or fuse with the cell's membrane to release the contents outside of the cell.

Membrane fusion is carried out by a group of proteins called SNAREs. These proteins are embedded on the membranes of both the vesicle and its target, and they bind to each other to form a tight complex. This complex docks the vesicle to the target and then acts like a "zipper" to pull the two membranes close enough to fuse. The best-studied SNARE proteins act in nerve cells and fuse vesicles to the cell's membrane in order to release molecules called neurotransmitters. This process is essential for communication between nerve cells, and relies on a protein called Munc18-1. However, it is not well understood how SNARE proteins assemble into the complex and how Munc18-1 regulates this process.

Ma et al. have now used a tool called "optical tweezers" to pull an assembled SNARE complex apart in the laboratory and then observe how it folds and assembles in a step-by-step process. These experiments showed that the complex assembled in four stages and not three as has been reported in previous work. SNARE proteins are made up of four parts called domains, and Ma et al. observed that the N-terminal domains were the first to bind to each other. Next, the binding progressed to the middle domain, then to the C-terminal domain and finally to the linker domain. An intermediate, half-zippered form was also observed.

Ma et al. next analysed each domain in more detail and found that the N-terminal and C-terminal domains drive the docking of vesicles to the target membrane, the middle domain is crucial for assembling the SNARE complex correctly, and all three domains regulate the fusing of the membranes. Further experiments showed that Munc18-1 promoted the assembly of new SNARE complexes and stabilized the half-zippered form, rather than stabilizing the complex after it had fully assembled. This study will provide a new tool to examine many other proteins that regulate SNARE assembly, and a basis to understand the role of SNARE proteins in brain activity.

(*Figure 1B*). The tight SNARE association is mediated by 15 layers of hydrophobic amino acids and a central ionic layer in the core of the bundle (*Fasshauer et al., 1998*) (*Figure 1A*). Numerous studies have shown that layer mutations differentially impact synaptic exocytosis, brain functions, and human psychology (*Walter et al., 2010*; *Mohrmann et al., 2010*; *Shen et al., 2014*; *Sudhof, 2014*). Thus, it is crucial to understand how SNARE assembly dictates stage-wise exocytosis and how SNARE mutations alter the energetics and kinetics of SNARE assembly to cause their observed phenotypes.

The molecular mechanism by which Munc18-1 regulates SNARE assembly, although essential for membrane fusion, has long been a matter of debate (*Rizo and Sudhof, 2012*). On one hand, Munc18-1 can significantly enhance the rate and specificity of SNARE-mediated membrane fusion under certain conditions (*Shen et al., 2007*; *Rathore et al., 2010*; *Yu et al., 2015*). On the other hand, Munc18-1 tightly associates with syntaxin in a closed conformation that inhibits SNARE assembly (*Colbert et al., 2013*; *Burkhardt et al., 2008*; *Dulubova et al., 1999*). Munc18-1 also binds with over micromolar affinities to the t-SNARE complex (*Zhang et al., 2015*), VAMP2 (*Parisotto et al., 2014*; *Xu et al., 2010b*), and the ternary SNARE complex (*Dulubova et al., 2007*; *Shen et al., 2007*; *Rathore et al., 2010*). The exact functions of these associations in SNARE assembly and membrane fusion are not well understood (*Shen et al., 2007*; *Burkhardt et al., 2008*; *Zhou et al., 2013*).

SNAREs couple their exergonic folding to membrane fusion (*Sudhof and Rothman, 2009*; *Rothman, 2014*). Such a thermodynamic coupling mechanism predicts that any perturbation in SNARE assembly impacts membrane fusion. Significant progresses have been made to test the prediction in the past two decades (*Chen et al., 1999*; *Walter et al., 2010*; *Mohrmann et al., 2010*). However, a quantitative test requires accurate and comprehensive measurements of both the energies and kinetics of SNARE assembly and the rates of membrane fusion for wild-type and various mutant SNARE complexes. Whereas SNARE-mediated membrane fusion has been widely examined in vitro

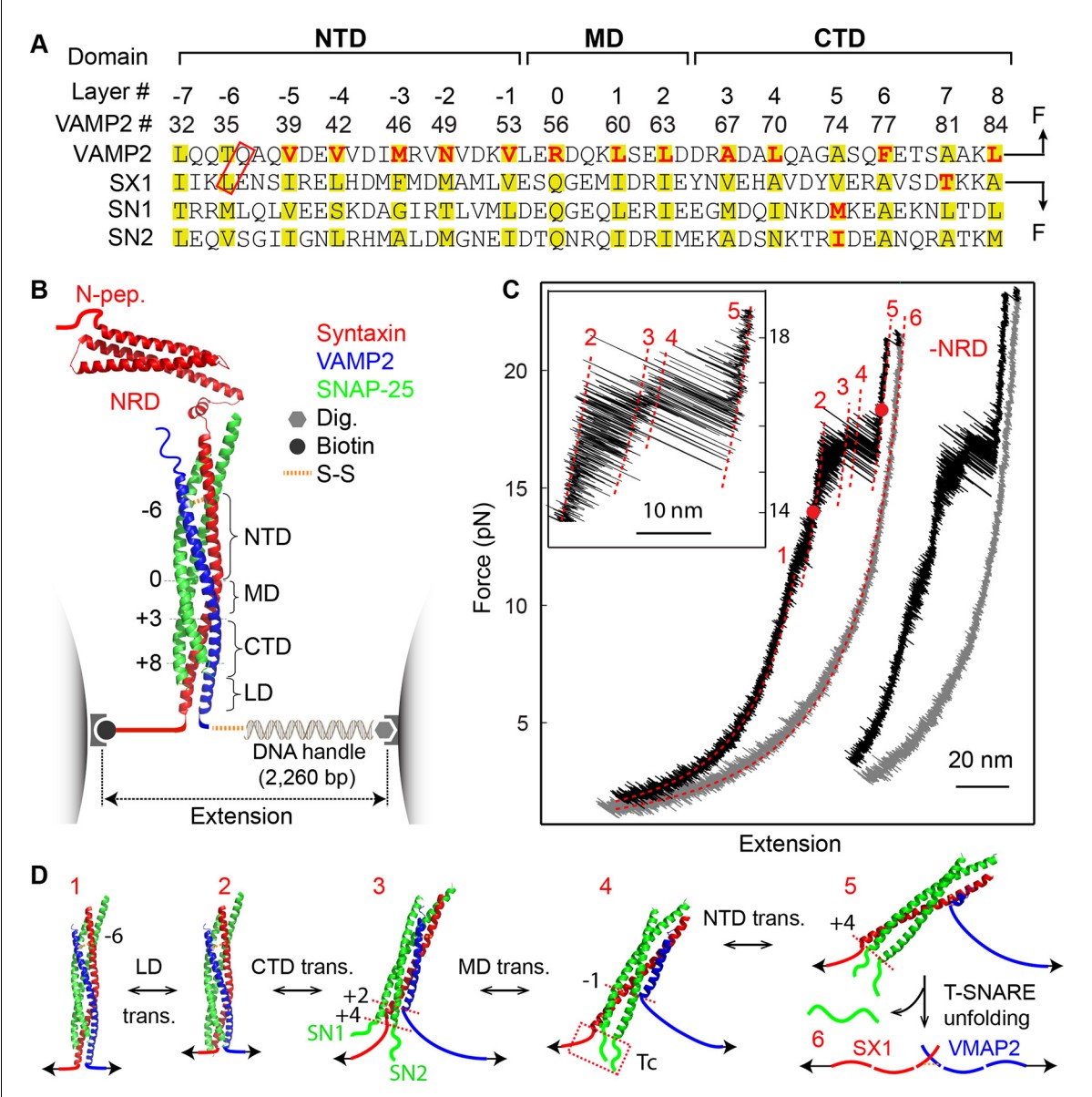

**Figure 1.** Four distinct stages of SNARE assembly. (**A**) Sequences and domain structures of SNARE motifs in VAMP2, syntaxin 1A (SX1), and SNAP-25B (SN1 and SN2). The amino acids in different layers are highlighted in yellow, with the corresponding layer numbers and VAMP2 amino acid numbers labeled. The two amino acids in the red rectangle were mutated to cysteine and crosslinked by a disulfide bridge. The layer amino acids mutated in this study are highlighted in red. (**B**) Diagram of the experimental setup (*Gao et al., 2012*; *Cecconi et al., 2005*). Different functional domains of the SNARE complex are indicated, including the N-terminal domain (NTD), the middle domain (MD), the C-terminal domain (CTD), the linker domain (LD), as well as the N-terminal regulatory domain (NRD) in syntaxin. The positions of four bordering layers are indicated by their layer numbers. (**C**) Force-extension curves (FECs) of a single SNARE complex with or without (-NRD) the NRD obtained by pulling (black) and then relaxing (gray) the complex. The FECs of the two complexes generally overlap but were shifted along x-axis for clarity. The continuous regions of the FECs corresponding to different assembly states (marked by red numbers, see D) were fitted by the worm-like chain model (red lines). The inset shows a close-up view of the region marked by two red dots. Throughout this work, the time-dependent extension and force were mean-filtered using a time window of 5 ms and plotted in the FECs shown, if not otherwise specified. (**D**) Six different SNARE assembly states. The states are numbered the same throughout the text (in red). Black numbers indicate different layers. The disordered t-SNARE C-terminus (Tc) in all partially zippered and unzipped SNARE complexes is marked by a dashed rectangle in state 4. The NRD minimally affects intrinsic SNARE assembly and is omitted in the SNARE structures depicted here.

The following figure supplements are available for figure 1:

**Figure supplement 1.** N-terminal amino acid sequences of rat VAMP 2 and syntaxin-1A showing their different crosslinking sites (marked by red or green boxes) in the SNARE complexes tested.

*Figure 1 continued on next page*

*Figure 1 continued*

**Figure supplement 2.** Force-extension curves (FECs) of the SNARE complexes crosslinked at different sites shown in *Figure 1—figure supplement 1*.

**Figure supplement 3.** FECs of the SNARE complex showing reversible and irreversible transitions and effects of crosslinking near the -6 layer.

**Figure supplement 4.** FECs of a single SNARE complex obtained in the presence 15 μM SNAP-25 in the solution showed that SNAP-25 was required for SNARE assembly.

**Figure supplement 5.** Time-dependent extension (top), instantaneous force (middle), and trap separation (bottom) of the SNARE construct VI (green) or I (black) as the SNARE complex was being pulled by increasing the trap separation at a speed of 10 nm/s.

and in vivo, the energetics and kinetics of SNARE assembly have not been well characterized using traditional experimental approaches (*Fasshauer et al., 2002*; *Wiederhold and Fasshauer, 2009*). Consequently, a quantitative Munc18-1-dependent model linking the thermodynamics of SNARE assembly to membrane fusion is lacking. For example, although VAMP2 layer mutation F77A abolished exocytosis, isothermal titration calorimetry (ITC) measurements showed that the mutation barely changed SNARE folding energy (*Walter et al., 2010*). To address this challenge, we have recently established a single-molecule assay for SNARE assembly based on high-resolution optical tweezers and characterized the folding intermediates, energies, and kinetics of four representative SNARE complexes (*Gao et al., 2012*; *Zorman et al., 2014*). We have found that all the SNARE complexes are similarly unzipped in two reversible steps and one irreversible step, corresponding to sequential disassembly of SNARE LD, CTD, and NTD at increasing forces. However, complete stage-wise *assembly* has not directly been observed, due to the slow NTD association. Here, we developed a new SNARE construct that allowed us to measure reversible SNARE assembly and disassembly in four distinct stages and to determine their folding energies and kinetics (*Figure 1B*). We found that these different assembly stages are regulated by Munc18-1 and play distinct roles in membrane fusion.

## Results

### Single-molecule manipulation of the SNARE complex

We previously studied a truncated cytoplasmic domain of the synaptic SNARE complex lacking the N-terminal regulatory domain (NRD) of syntaxin (*Gao et al., 2012*). The NRD contains a ~15 amino acid (a.a.) N-terminal peptide (N-peptide) and a C-terminal $H_{abc}$ domain in a three-helix bundle conformation (*Fernandez et al., 1998*) (*Figure 1B*). To observe reversible and regulatory SNARE assembly, we designed SNARE complexes containing the full cytoplasmic domain and an N-terminal crosslinking site between syntaxin and VAMP2. We examined six such SNARE constructs crosslinked at different sites (*Figure 1—figure supplements 1,2*) and found that the construct crosslinked near the -6 hydrophobic layer exhibited fast and reversible NTD folding/unfolding transition (*Figure 1A, B,C*). Compared to the slow and irreversible NTD association detected for the SNARE constructs crosslinked N-terminal to the -7 layer (*Gao et al., 2012*) (*Figure 1—figure supplements 1–2*, construct I), the fast NTD folding observed here implicated the existence of a nucleation site for NTD zippering near or N-terminal to the -6 layer. This new construct was used for most of the experiments described below. To manipulate a single SNARE complex, we either pulled or relaxed the complex by moving one optical trap relative to the other at a speed of 10 nm/s or held the complex under constant average force at a fixed trap separation (*Figure 1B*). Both force and extension of the protein-DNA tether were recorded at 10 kHz and used to derive the conformational and energetic changes of the SNARE complex in real time (*Gao et al., 2012*). Specifically, for a reversible two-state transition, the folding energy of the associated protein domain can be measured based on the mechanical work required to unfold the domain, which is equal to the equilibrium force multiplied by the extension change related to the transition (*Bustamante et al., 2004*; *Liphardt et al., 2001*; *Gao et al., 2011*).

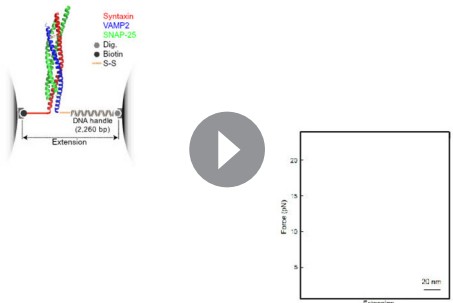

**Video 1.** Optical tweezers manipulate a single SNARE complex to reveal its folding intermediates, energetics, and kinetics. Not drawn to scale. Note that the single-SNARE complex was being pulled or relaxed by moving the optical trap on the right (not shown) at a speed of 10 nm/s. The tension and extension of the DNA-protein tether was monitored to derive the conformations, energetics, and kinetics of SNARE assembly.

## Four stages of SNARE assembly in four domains

The force-extension curves (FECs) obtained by pulling and then relaxing a single SNARE complex (*Figure 1C*) showed four reversible transitions with fast extension flickering (among states 1 to 5) and one irreversible unfolding transition (from states 5 to 6), indicating cooperative unfolding/refolding of five SNARE domains (*Video 1*). These discrete transitions were separated by continuous regions caused by elastic response of the SNARE-DNA tether to force change, while the SNARE complex remained in the same folding state. Accordingly, these FEC regions could be fit by a worm-like chain model for both the DNA handle and the unfolded polypeptide (*Bustamante et al., 1994*; *Marko and Siggia, 1995*). The fitting of these regions yielded different contour lengths for the polypeptide unfolded in the SNARE complex, which were used to derive the structures of SNARE assembly intermediates (*Figure 1D*). The FECs associated with states 1–3 were identical to the FECs previously reported (*Gao et al., 2012*) and independent of the N-terminal crosslinking (*Figure 1—figure supplement 2,3*). The comparison suggests that the states 1, 2, and 3 are the fully folded, the LD-unfolded, and the partially-zippered SNARE complexes, respectively (*Figure 1D*). A new reversible NTD transition partly overlapped the CTD transition (between states 3 and 5, *Figure 1C*, inset). Relaxing the complex from state 5 led to a FEC that overlaps the FEC in the pulling phase, indicating complete and reversible reassembly of the SNARE complex (*Figure 1—figure supplement 3*). However, pulling the complex in state 5 to higher forces saw a small rip resulting from unfolding of the remaining t-SNARE complex (from state 5 to state 6) (*Gao et al., 2012*) (*Figure 1C,D*). No additional unfolding event was observed when further pulling the complex in state 6 (*Figure 1—figure supplement 3*, #3), indicating that the SNARE complex was fully disassembled. We then relaxed the molecule to a force of 1 pN to check if SNARE reassembled. Unexpectedly, no single SNARE re-assembly event was detected among 45 different SNARE complexes tested. Consistent with our previous observation (*Gao et al., 2012*), the failure of reassembly was caused by dissociation of the SNAP-25 molecule upon t-SNARE unfolding, because adding 15 μM SNAP-25 into the solution rescued the assembly (*Figure 1—figure supplement 4*).

Close inspection shows an additional state hidden in the NTD transition, state 4 (*Figure 1—figure supplement 5*). This state had a maximum lifetime of ~1 ms and rapidly transited between state 3 and state 5 (*Figure 2A* and *Figure 2—figure supplement 1*). This transition was clearly seen in some FECs and extension traces plotted at ≥1 kHz bandwidth (*Figure 2A*) and was also found in our previous measurements (*Gao et al., 2012*) (*Figure 1—figure supplement 5*). Based on the measured extension changes accompanying the state transitions and the well-defined structures of the initial and final states of SNARE assembly (states 1 and 6), we derived the intermediate conformations (*Figure 1D* and 'Materials and methods'). In the partially-zippered state 3, the VAMP2 CTD and the t-SNARE CTD are unfolded from layer +8 to layer +3 and layer +5, respectively, whereas the remainder of the SNAREs are largely helical as in the four-helix bundle. This structure is consistent with that of a different trans-SNARE complex located on the yeast vacuole (*Schwartz and Merz, 2009*). In the MD-unfolded state 4, VAMP2 is unfolded further to the ionic layer. Therefore, the MD consists of the ionic layer (0 layer) and the +1 and +2 layers exclusively comprising leucine and isoleucine residues known to form strong coiled coils (*Figure 1A*), which leads to the higher mechanical stability of the MD than the CTD and their distinct transitions. In the unzipped SNARE state 5, the t-SNARE complex remains in a three-helix bundle conformation, but with a frayed C-terminus (Tc). These conformations were further corroborated by results of protein binding and SNARE mutations described below.

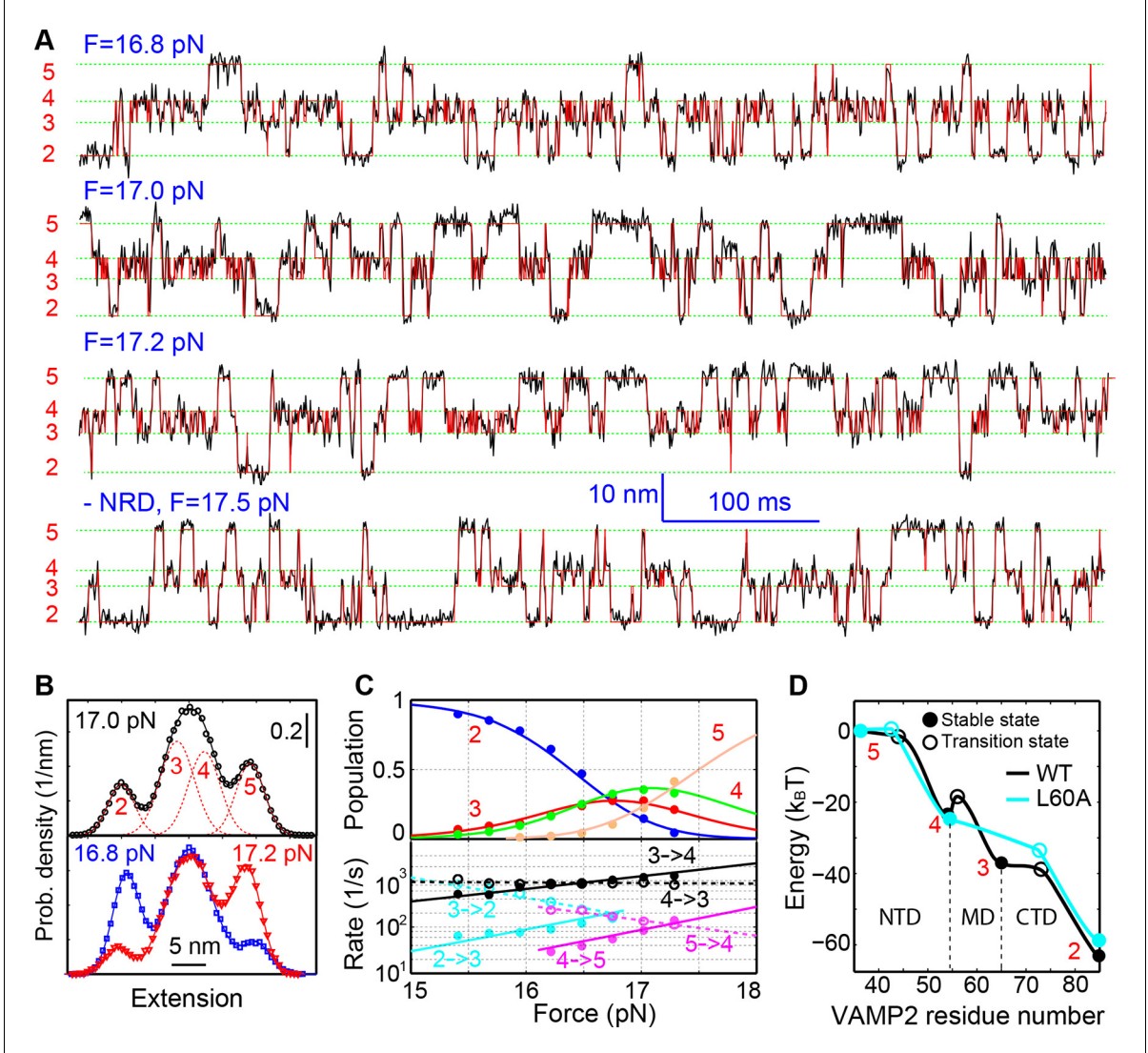

**Figure 2.** Energetics and kinetics of SNARE assembly. (**A**) Extension-time trajectories of a single SNARE complex under the indicated constant mean forces. The idealized state transitions derived from hidden-Markov modeling (HMM) are shown as red lines. The positions of different states (red numbers) are marked by green dashed lines. Data were mean-filtered using a time window of 0.2 ms and shown. (**B**) Probability density distributions of the extensions corresponding to the first three traces in A and their best fits by a sum of four Gaussian functions (lines). (**C**) HMM-derived state populations (upper panel, symbols) and transition rates (lower panel, symbols) are shown as a function of the mean forces. The unfolding and folding rates are shown as solid and hollow symbols, respectively. Their best model fits are shown as lines ('Materials and methods'). (**D**) Simplified folding energy landscapes of the wild-type (WT) and the mutant (L60A) SNARE complexes. The abscissa denotes the VAMP2 residue number bordering the structured and unstructured polypeptide regions in the corresponding folding state. All stable states (solid) and transition states (hollow) are defined in the presence of forces.

The following figure supplements are available for figure 2:

**Figure supplement 1.** Force-dependent average lifetimes of the four states 2–5 involved in the SNARE assembly (symbols) and their best fits (lines).

**Figure supplement 2.** Transition rates determined by hidden-Markov modeling (HMM) shows that the t- and v-SNAREs sequentially assemble like a molecular zipper.

**Figure supplement 3.** Modeling SNARE assembly observed in optical traps using the crystal structure of the SNARE complex and a simplified energy landscape.

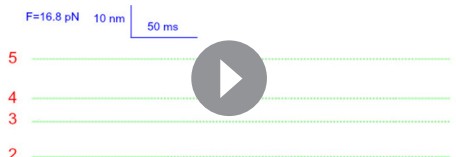

**Video 2.** Force-induced SNARE folding and unfolding transitions under thermodynamic equilibrium among four distinct states. The single SNARE complex was held at the indicated mean force at a constant trap separation. The stochastic unfolding and refolding transitions of the SNARE complex represent its thermodynamic fluctuations when the free energies of the four states become close (<5 $k_BT$) in the presence of force (*Liphardt et al., 2001*).

In conclusion, we identified an independent SNARE folding domain, the middle domain, and observed that the SNARE complex zippers in four distinct and sequential stages from NTD to MD to CTD and finally to LD. These results clarify the partially zippered SNARE states previously derived (*Schwartz and Merz, 2009*; *Gao et al., 2012*; *Min et al., 2013*).

## Energetics and kinetics of SNARE assembly

To better characterize the intermediates and their associated energies and transition kinetics, we measured the extension-time trajectories of the SNARE complex at different constant mean forces. Simultaneous transitions among four states 2–5 could be resolved (*Figure 2A* and *Video 2*). The positions, populations, transition rates, and lifetimes (*Figure 2—figure supplement 1*) associated with these states could be determined by hidden-Markov modeling (HMM) (*McKinney et al., 2006*; *Gao et al., 2011*; *Rebane, et al., 2016*). Accordingly, the probability density distributions of the extensions were fit well by a sum of four Gaussian functions (*Figure 2B*), confirming the four-state transitions and revealing the average state positions and populations consistent with those derived from HMM. *Figure 2C* shows populations of the four states (top panel) and rates of the main transitions among these states (bottom panel), measured on a single representative SNARE complex. The transitions are approximately sequential, because the rates for sequential transitions are over two orders of magnitude greater than the rates for non-sequential transitions (*Figure 2—figure supplement 2*).

We developed a nonlinear model to describe protein transitions in optical traps (*Rebane, et al., 2016*; *Gao et al., 2012*; *Xi et al., 2012*) (*Figure 2—figure supplement 3* and 'Materials and methods'). We modeled SNARE zippering as VAMP2 folding along a t-SNARE template based on the crystal structure of the SNARE complex. The model included the extension and energy contributions of both the folded and unfolded portions of the complex along the pulling direction. In addition, we calculated the populations of different folding states and their transition rates based on the model. Simultaneous nonlinear least-squares fitting of the calculations to the corresponding measurements yielded the conformations (*Figure 1D*) and energies of the three intermediate states and their associated transition states at zero force, resulting in the simplified energy landscape of SNARE folding/assembly (*Figure 2D*). Specifically, the folding energies of NTD, MD, and CTD are 25 (±2, S.D.) $k_BT$, 13 (±1) $k_BT$, and 22 (±3) $k_BT$, respectively, where $k_B$ is the Boltzmann constant and T= 296 K the room temperature, with $k_BT$=0.59 kcal/mol. Thus, the SNARE complex has total zippering energy of 68 (±4) $k_BT$, including 8 (±2) $k_BT$ LD folding energy (*Gao et al., 2012*). Folding of the NTD and CTD is downhill. In contrast, the MD has a folding energy barrier of 5.0 (±0.6) $k_BT$ located close to the ionic layer, indicating a high-energy penalty to dehydrate the ionic layer for MD assembly (*Rebane, et al., 2016*).

## Vc peptide induces folding of t-SNARE C-terminus and attenuates SNARE zippering

A distinct feature of our derived conformations is a frayed Tc in states 3–5 (*Figure 1D*). To confirm the disordered Tc, we repeated the above experiments by adding a peptide comprising the C-terminal sequence 49–96 or layers from −2 to +8 of VAMP2 (Vc peptide) to the solution (*Figure 1A*). We predicted that binding of the Vc peptide to the t-SNARE complex would induce a coil-to-helix transition in Tc and inhibit or attenuate SNARE zippering into the two partially-zippered states 3 and 4. The Vc peptide is widely used to facilitate studies of SNARE-mediated fusion (*Melia et al., 2002*; *Pobbati et al., 2006*; *Hernandez et al., 2012*). Pre-bound to the t-SNARE complex, this peptide greatly enhances membrane fusion, probably because the Vc peptide stabilizes the t-SNARE complex in a conformation ready to pair with VAMP2, and thus helping to initiate SNARE zippering

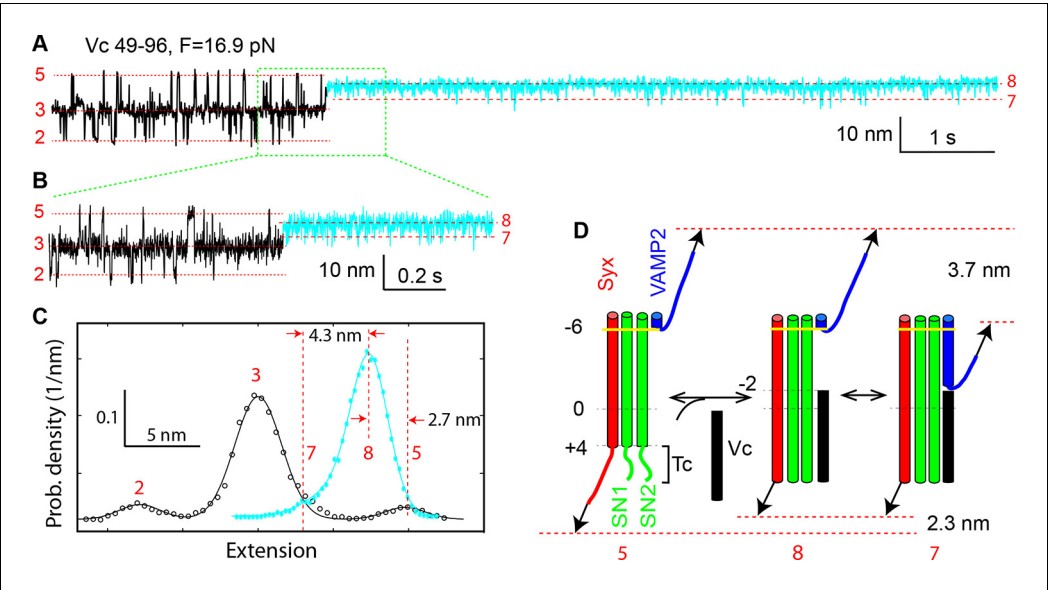

**Figure 3.** Vc peptide induces a coil-to-helix transition in the t-SNARE C-terminus (Tc). (**A**) Extension-time trajectory showing changes in SNARE folding kinetics caused by binding of the Vc peptide to the t-SNARE in the SNARE complex. The Vc-bound region is colored in cyan. The positions of different states are marked by red dashed lines and the corresponding state numbers. Data were filtered using a time window of 1 ms. (**B**) Close-up view of the region in A marked by a rectangle. Data were filtered using a time window of 0.6 ms. (**C**) Probability density distributions of the extensions in A before (black) and after (cyan) Vc binding. (**D**) Diagram illustrating the Vc-induced conformational transitions. The extension changes averaged over eight Vc-binding events are indicated.

(*Pobbati et al., 2006*). In the presence of 44 µM Vc peptide in the solution, we found that the Vc peptide frequently bound to the SNARE complex. The binding was manifested by a sudden SNARE transition into new Vc-bound states (cyan region in *Figure 3A,B*). Correspondingly, the binding caused a dramatic change in the histogram distribution of the extension (*Figure 3C*). Once bound, the Vc peptide did not dissociate from the SNARE complex, indicating that the Vc peptide outcompeted the VAMP2 molecule to tightly associate with the t-SNARE complex under our experimental conditions. The lifetime of the Vc-bound states was estimated to be greater than 10 min in the force range of NTD and CTD transitions. The Vc-bound SNARE states have a unimodal extension distribution (*Figure 3C*), whose peak extension is 2.7 nm less than the average extension of the NTD-unfolded state 5, with an average shortening of 2.3 (±0.5, N=8) nm (*Figure 3D*). The extension shortening upon Vc binding is consistent with the predicted coil-to-helix transition in Tc, confirming the disordered Tc. The comparison of the extension distributions also reveals that the Vc peptide predominantly stabilized the SNARE complex in the Vn-unfolded state (state 8) (*Figure 3C*). Partially zippered states were only transient, indicated by downward extension excursions to a new state (state 7) with its extension greater than those of the partially zippered states 3 and 4. Thus, upon binding to the t-SNARE complex, the Vc peptide acted as a roadblock to attenuate VAMP2 from zippering. However, the Vc peptide was displaced by the VAMP2 molecule at an average force of 6 (± 1) pN upon relaxation. Taken together, these observations supported the conformations of the partially zippered states and the unzipped state derived by us (*Figure 1D*) and indicated that the Vc peptide strongly attenuated SNARE zippering.

## Position-dependent effects of layer mutation on SNARE assembly

To understand functions of the different stages of SNARE assembly in membrane fusion and exocytosis, we measured the folding energies and kinetics of 15 SNARE complexes mutated in each layer from −5 to +8. Their effects on exocytosis will be discussed in the forthcoming section. Most of these mutations are alanine substitutions in VAMP2 (*Walter et al., 2010*) (*Figure 1A*). Three other mutations are VAMP2 A67S, SNAP-25 M71A/I192A (*Mohrmann et al., 2010*), and syntaxin T251I

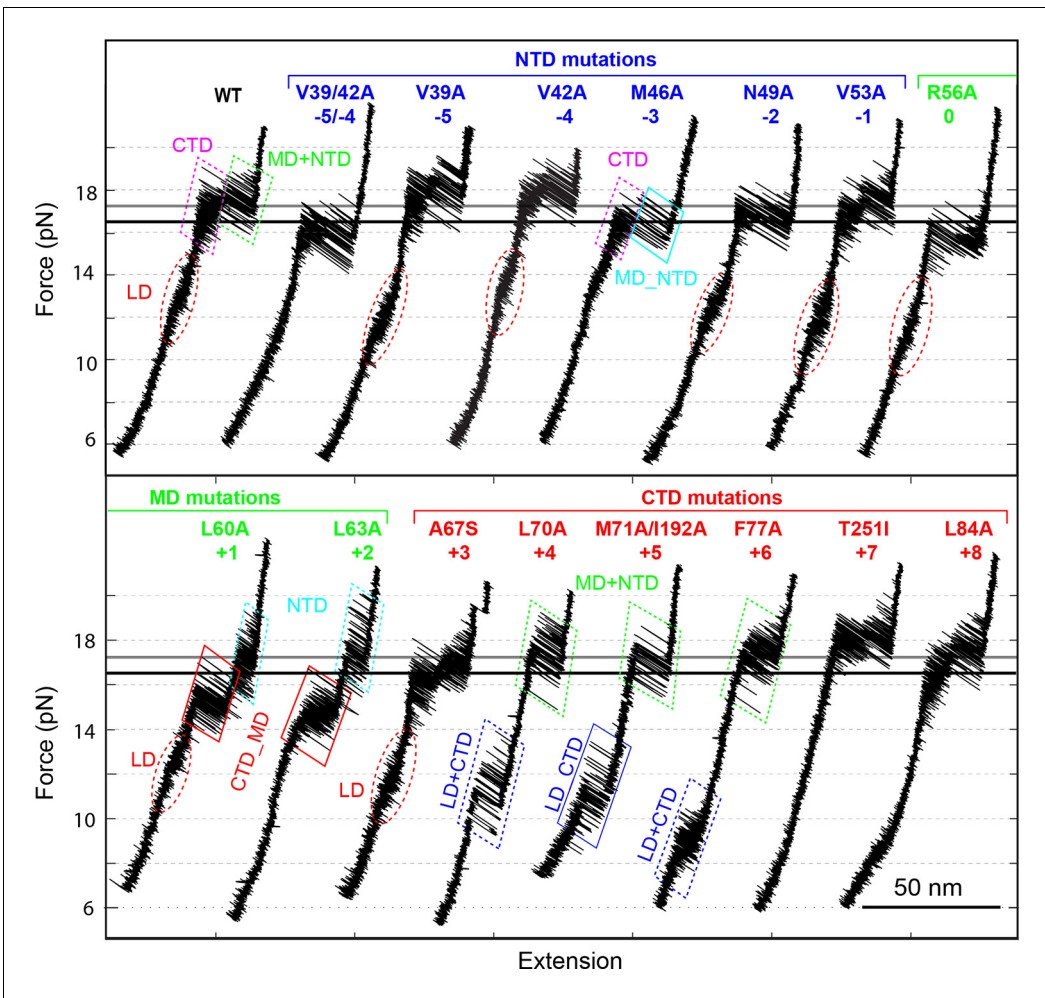

**Figure 4.** Position-dependent effects of layer mutations on SNARE assembly shown by the FECs of the wild-type and mutant SNARE complexes. The SNARE mutations and their layer numbers are colored based on their associated domains: blue for NTD, green for MD, and red for CTD. While most mutants assemble sequentially as the wild type (WT), some mutants exhibit altered folding pathways, in which two neighboring domains fold and unfold cooperatively as combined domains. Throughout the text, two-state transitions of these new domains are marked by their associated subdomains connected by "_" (*Figure 4—figure supplement 1*). By comparison, overlapping sequential transitions are indicated by their associated domains linked by "+". Different folding domains are marked by unique colored ovals or rectangles. Folding energies of the same domain in different SNARE complexes can be compared by the equilibrium forces of the domain transition. The black and gray horizontal lines indicate the average equilibrium forces of CTD (16.5 ± 0.8 pN) and MD+NTD (17.2 ± 0.8 pN), respectively, of the wild-type SNARE complex. CTD, C-terminal domain; FECs, Force-extensioncurves; MD, middle domain; NTD, N-terminal domain.

The following figure supplement is available for figure 4:

**Figure supplement 1.** Comparison of sequential and cooperative domain transitions, using MD and NTD as an example.

(*Lagow et al., 2007*). These mutations were expected to mainly perturb SNARE zippering but minimally interfere with binding of regulatory proteins to SNARE proteins, which allows us to focus on the effect of intrinsic SNARE assembly on exocytosis. Furthermore, most of the mutants have been characterized in vivo for their impacts in exocytosis and neurotransmitter release (*Walter et al., 2010*; *Mohrmann et al., 2010*; *Lagow et al., 2007*). Related mutations have been associated with many neurological disorders in humans (*Shen et al., 2014*; *Rohena et al., 2013*).

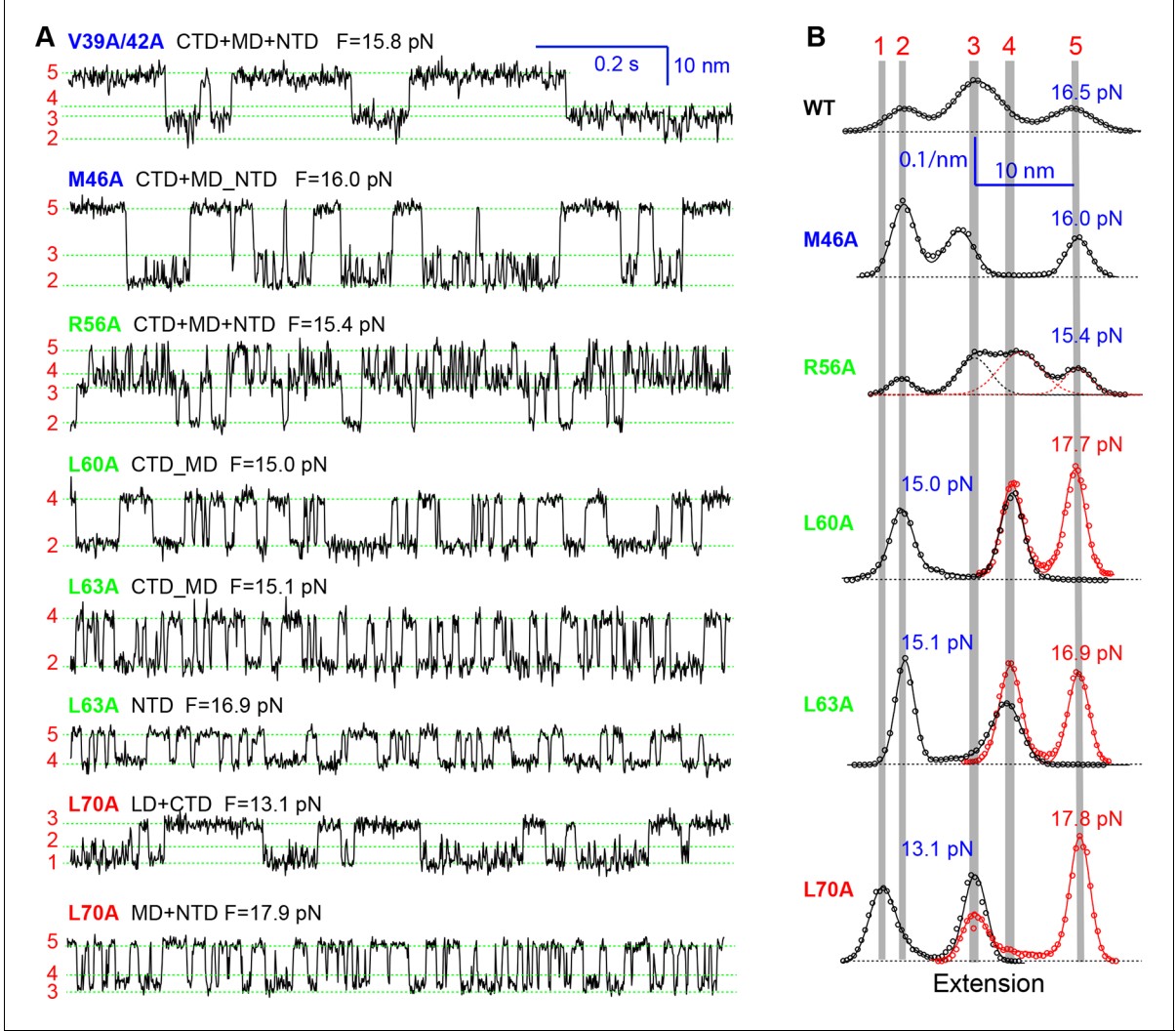

**Figure 5.** Extension-time trajectories of the mutant SNARE complexes under constant forces showing effects of mutations on the transition kinetics of different SNARE domains. (**A**) Extension-time trajectories. The domains involved in the observed transitions and the mean forces (F) are indicated following the colored mutation names in bold (*Figure 4—figure supplement 1*). Positions of different states are marked by green dashed lines. (**B**) Probability density distributions of the extensions under the indicated constant forces (symbols) revealing structural changes in the intermediates of SNARE assembly. The distributions could be fitted by 2-4 Gaussian functions (solid lines), and were horizontally shifted to align the peaks corresponding to different states (indicated by the vertical shaded bars).

The following figure supplements are available for figure 5:

**Figure supplement 1.** Extension-time trajectories of mutant SNARE complexes showing indicated domain transitions under constant mean forces.

**Figure supplement 2.** Probability density distributions of the extensions of the mutant SNARE complex R56A under different constant mean forces.

All SNARE complexes containing single alanine substitutions in the NTD, except for M46A, exhibit similar FECs and transition kinetics as the wild-type (*Figures 4–6* and *Figure 5—figure supplement 1*). Therefore, single alanine substitutions in the NTD generally have minor impact on SNARE assembly. In contrast, M46A reduced the stability of NTD, as is indicated by a small decrease in the force for NTD transition compared to the wild-type (*Figures 4* and *6*). The approximately equal mechanical stabilities of MD and NTD now led to disappearance of the intermediate state 4 (*Figure 5*). Consequently, both MD and NTD transited cooperatively as one unit (designated as MD_NTD) between the CTD-unfolded state 3 and the unzipped state 5 (*Figure 4—figure supplement 1*), in contrast with the sequential MD and NTD transitions (designated as MD+NTD) seen for

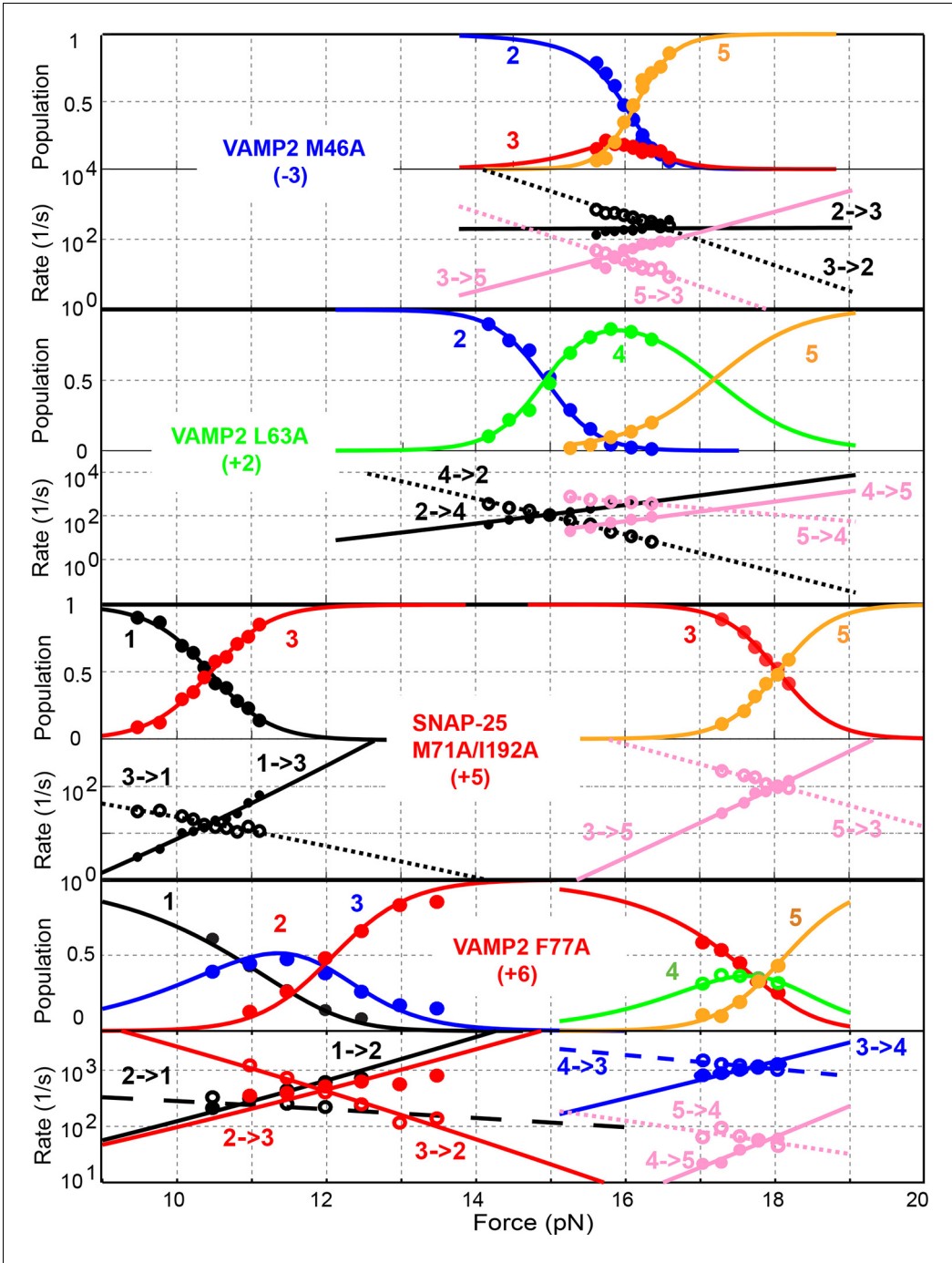

**Figure 6.** Comparison of force-dependent state populations and transition rates of the mutant SNARE complexes. Experimental measurements from representative single SNARE complexes (symbols) were fit by a simplified energy landscape model of SNARE assembly (lines) to derive the folding energies of different domains ('Materials and methods').

the wild-type SNARE complex (**Figure 2A**). Thus, M46A altered SNARE assembly kinetics and pathway. Finally, the double mutation V39A/V42A dramatically decreased the folding rate of NTD (**Figure 5**). In addition, it destabilized NTD by 3 $k_BT$ (**Figure 7**).

Three point mutations in MD caused great changes in SNARE zippering. First, R56A destabilized both NTD and CTD (**Figures 4** and **7**), indicating that the ionic layer globally stabilizes the SNARE

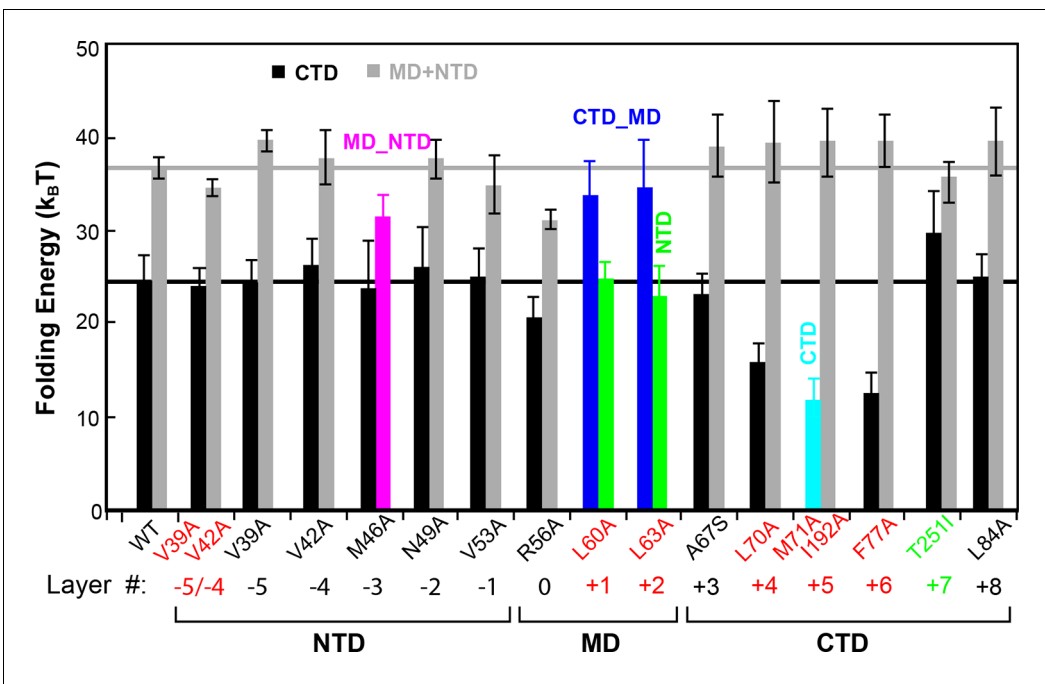

**Figure 7.** Layer mutations differentially affect folding energies and pathways of SNARE assembly. The wild-type and most mutant SNARE complexes assemble sequentially in the order of NTD, MD, CTD, and LD, with their total MD and NTD (MD+NTD) energies and CTD energies shown in black and gray bars, respectively. The energies of CTD and MD+NTD of the wild-type SNARE complex are marked by black and gray lines, respectively. Other mutants contain cooperative transitions involving two domains (*Figure 4—figure supplement 1*), which are indicated by colored bars. The double mutations in the +5 layer in SNAP-25 cause cooperative LD and CTD transition, but only the CTD energy is shown for better comparison, assuming an LD folding energy of 8 $k_BT$ (*Gao et al., 2012*). The mutant amino acids and their layer numbers that impair or enhance exocytosis are colored in red or green, respectively. Error bars represent standard deviations of the measurement. CTD, C-terminal domain; LD, linker domain; MD, middle domain; NTD, N-terminal domain.

complex (*Rebane, et al., 2016*). In addition, R56A increased the extension change associated with the MD transition (*Figure 5* and *Figure 5—figure supplement 2*), corroborating the independent MD transition and the N-terminal border position of R56 in the MD. In contrast, L60A (+1) and L63A (+2) abolished the independent MD transition and caused MD to merge with CTD (*Figure 5*). Consequently, the combined CTD and MD (CTD_MD) transited cooperatively as a distinct domain with an average extension change approximately equal to the sum (~11 nm) of the extension changes for the CTD (6–7 nm) and the MD (3–4 nm). Therefore, mutations at +1 and +2 layers altered the pathway of SNARE assembly, in which the distinct MD assembly was bypassed. Accordingly, all MD mutations abolished the energy barrier in MD folding (*Rebane, et al., 2016*) (*Figure 2D*). Finally, we found that all MD mutants frequently and abruptly switched their transition kinetics (*Figure 8*), indicating an important role of MD in accurate and robust SNARE assembly. Because of the abundant heterogeneity, the folding energies of the MD mutants were only measured from the canonical regions. In summary, the middle domain plays a pivotal role in robust and correct stage-wise SNARE assembly, and its mutation leads to changes in SNARE assembly pathway and accuracy.

The CTD layer mutations affected CTD and LD folding in a position-dependent manner, but did not alter NTD folding. The two mutations A67S (+3) and L84A (+8) at the borders of the CTD had only minor effects on SNARE assembly (*Figures 4*, *7*, and *Figure 5—figure supplement 1*). In contrast, L70A (+4) and F77A (+6) in VAMP2 and M71A/I192A (+5) in SNAP-25 greatly reduced the force ranges of the CTD transition that now overlapped the LD transition. In particular, the +6 layer mutation weakened both CTD and LD (*Figures 4,6,7*, and *Figure 5—figure supplement 1*). Correspondingly, alanine substitutions at +4, +5 and +6 layers dramatically decreased CTD folding energies by 10 $k_BT$, 13 $k_BT$, and 12 $k_BT$, respectively, compared to the wild-type. Our energy

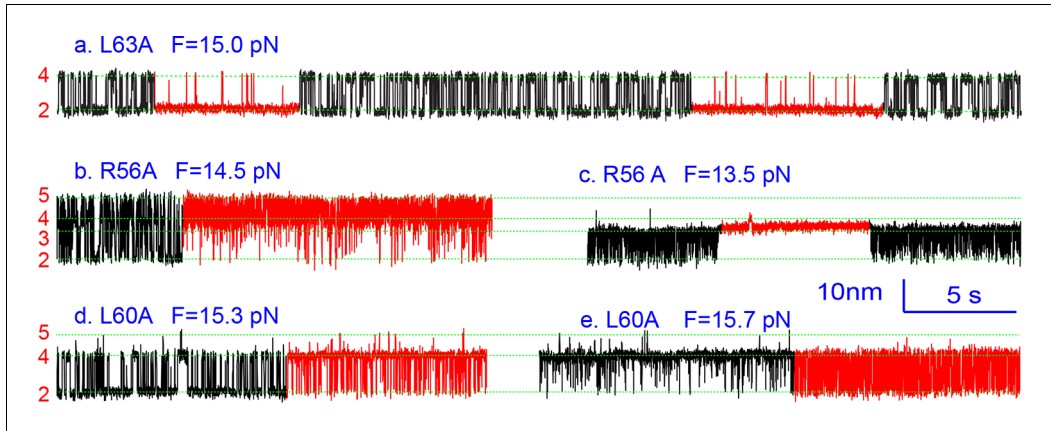

**Figure 8.** Extension-time trajectories showing kinetic heterogeneity in SNARE folding due to MD Layer Mutation. The canonical (occurring more frequently) and the altered transitions are shown in black and red, respectively. The heterogeneity in SNARE assembly was manifested by abrupt changes in transition rates and state populations. Some of these kinetic switching events (a and c) were reversible in our experimental time scale (typically <30 min), indicating that the kinetic heterogeneity was caused neither by impurities in our SNARE constructs nor by photodamage of SNARE proteins in optical traps. However, the underlying molecular mechanism for such heterogeneity is unknown, but likely related to the metastable t-SNARE complex.

measurement of F77A contrasts with the previous measurement by ITC and is consistent with its ablation of exocytosis (*Walter et al., 2010*). The syntaxin mutation T251I (+7) is unusual, because it increased the CTD zippering energy by 5 $k_B$T (*Figures 4* and *7*), quantitatively confirming the previous inference based on structural modeling (*Lagow et al., 2007*). Taken together, the CTD stability is sensitive to mutations in layers from +4 to +7.

In summary, the effects of layer mutations on SNARE assembly strongly depend on their positions, ranging from minimal perturbation to significant changes in the energetics, intermediates, or kinetics of SNARE assembly.

## Munc18-1 promotes de novo SNARE assembly

To examine effects of Munc18-1 on SNARE assembly, we added 2–50 µM Munc18-1 into the solution where a single SNARE complex (*Figure 1B*) was being pulled in a microfluidic chamber (*Zhang et al., 2012*). We observed three force-dependent activities of Munc18-1 in SNARE assembly, consistent with the previously reported multiple modes of binding between Munc18-1 and the SNARE complex (*Sudhof and Rothman, 2009*; *Sudhof, 2014*).

The most distinct Munc18-1 activity is to enhance de novo SNARE assembly. To observe this activity, we completely unfolded the SNARE complex and then relaxed the complex to a low force to detect its possible reassembly (*Figure 9A*). In the absence of Munc18-1, no SNARE reassembly was observed (*Figure 1C* and *9B*) due to dissociation of the SNAP-25 molecule (*Figure 1—figure supplement 4*). In contrast, in the presence of 10 µM Munc18-1, 45% of 113 completely unfolded SNARE complexes showed full reassembly (*Figure 9A,B*). This observation suggests that Munc18-1 held the SNAP-25 molecule together with the tethered syntaxin or VAMP2 as t-SNARE unfolded, facilitating de novo SNARE assembly. The SNARE assembly occurred in a force range of 1-16 pN, with an average force of 8.5 pN (*Figure 9C*). In addition, the assembly events tended to appear in consecutive rounds of pulling and relaxing the same SNARE complex (*Figure 9A*, #3-#5). Although the probability to observe a SNARE reassembly event was 0.45 under our experimental conditions, the probability to detect a SNARE reassembly event after another was 0.8, suggesting a strong correlation between the different reassembly events mediated by Munc18-1. An appealing interpretation for the correlation is that the same Munc18-1 molecule mediated the consecutive reassembly events without dissociation from the SNARE complex during its multiple rounds of disassembly and reassembly. The Munc18-1-mediated SNARE assembly was not abolished by the two VAMP2 layer mutations L60A and L63A (*Figure 9B*). This finding suggests that, despite their effect on the

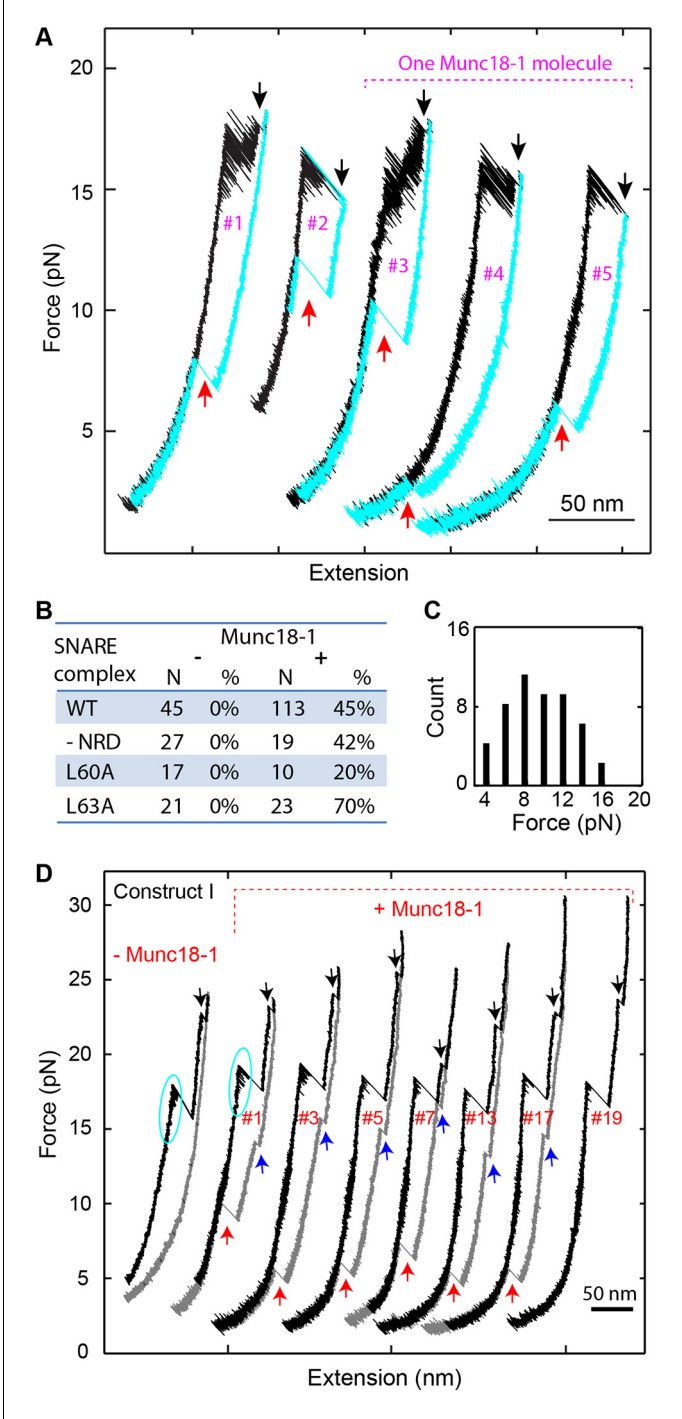

**Figure 9.** Munc18-1 helps to initiate SNARE assembly. (**A**) FECs of single SNARE complexes obtained in the presence of 10 µM Munc18-1 in the solution. Black and red arrows indicate t-SNARE unfolding and de novo SNARE reassembly, respectively. The FECs #3-#5 were obtained by consecutively pulling and relaxing the same SNARE complex. (**B**) Percentage of de novo assembly (%) among a total number of unfolded SNARE complexes tested (N) for the wild-type (WT), NRD-removed (-NRD), and mutant (L60A and L63A) SNARE complexes in the presence (+) or absence (−) of 10 µM Munc18-1. (**C**) Histogram distributions of the forces associated with the Munc18-1-mediated de novoSNARE assembly. (**D**) FECs of the SNARE complexes crosslinked at a site N-terminal to the SNARE motifs (**Figure 1—figure supplement 1**, construct I) in the absence (−Munc18-1) or presence (+Munc18-1) of Munc18-1. In the absence of Munc18-1, SNAP-25 tended to dissociate from the tethered syntaxin and VAMP2 when the t-SNARE complex was unfolded (black arrows). As a result, the SNARE complex generally
*Figure 9 continued on next page*

*Figure 9 continued*

reassembled for no more than four rounds during repetitive pulling and relaxation (*Gao et al., 2012*). However, in the presence of 10 μM Munc18-1, the yield of SNARE reassembly was greatly enhanced, leading to robust SNARE disassembly and reassembly for at least 19 rounds on the single SNARE complex. For clarity, only the FECs of some rounds of pulling and relaxation (indicated by the red numbers) are shown. In this case, full SNARE reassembly (red arrows) was preceded by t-SNARE formation (blue arrows) in the force range of 12-17 pN, due to the slow and force-sensitive NTD assembly of this construct. FECs, force-extension curves; NTD, N-terminal domain.

accuracy of intrinsic SNARE assembly, neither MD mutation significantly affect Munc18-1-dependent de novo SNARE assembly. Finally, we found that Munc18-1 similarly promoted SNARE assembly (*Figure 9D*) when the SNARE complex was crosslinked at a site N-terminal to the SNARE bundle (Construct I in *Figure 1—figure supplement 1*), indicating that the Munc18-1 activity was not caused by specific SNARE crosslinking. Interestingly, in this case, SNARE assembly occurred in two steps: t-SNARE formation at 12-17 pN, followed by cooperative SNARE zippering at 5-10 pN. Thus, Munc18-1 may mediate de novo SNARE assembly via a t-SNARE intermediate.

## Munc18-1 stabilizes the half-zippered SNARE complex

Under constant forces, we observed a series of prominent long-dwelling states in the middle of fast SNARE transitions (*Figure 10A,B*), revealing a new Munc18-1-dependent state. The state lasted from 0.03 to ~30 s, with an average dwell time of 2.8 s (*Figure 10C*) and was detected in a force range of 15-19 pN, with an average of 17.1 pN (*Figure 10D*). The new state has an extension corresponding to VAMP2 unzipped to a position between layers 0 and +1 (*Figure 10E*), and is thus a half-zippered state (state 9). The occurrence rate of this state was 0.012 s$^{-1}$ on average measured from a total experimental time of 6908 s. The rate was strongly force-dependent (*Figure 10D*). As a result, the half-zippered state was not found in the force range below 15 pN.

To stabilize the SNARE complex in a half-zippered state, Munc18-1 needs to stabilize NTD and to destabilize CTD. As a step to pinpoint the mechanism, we repeated the above experiment using the SNARE complex containing the VAMP2 mutation L70A. We found that L70A did not significantly affect the Munc18-1-stabilized half-zippered state (*Figure 10A*, iii), implying that Munc18-1 did not strongly interact with L70 in this observed Munc18-1 activity. Instead, Munc18-1 might partly bind the C-terminal domain of the t-SNARE in the SNARE complex to attenuate CTD zippering.

In conclusion, Munc18-1 stabilized the half-zippered SNARE complex in a force-dependent manner. Because the trans-SNARE complex is likely half-zippered (*Gao et al., 2012*; *Kyoung et al., 2011*; *Min et al., 2013*; *Li et al., 2014*), our observation corroborates that Munc18-1 enhances assembly of trans-SNARE complexes to promote membrane fusion (*Shen et al., 2007*; *Rathore et al., 2010*).

## Munc18-1 requires NRD to stabilize the half-zippered SNARE complex

Previous experiments show that Munc18-1 requires the NRD to enhance membrane fusion in vitro and exocytosis in vivo (*Shen et al., 2007*; *Dulubova et al., 2007*; *Zhou et al., 2013*). Specifically, Munc18-1 is recruited by the N-peptide in the NRD to bind trans-SNARE complexes (*Shen et al., 2010*). Interestingly, once Munc18-1 associates with trans-SNARE complexes, the NRD becomes dispensable for subsequent membrane fusion (*Rathore et al., 2010*). These findings suggest that Munc18-1 binds SNARE complexes in both NRD-dependent and NRD-independent modes and can dynamically transit between the two binding modes (*Hu et al., 2011*). To pinpoint the possible role of the NRD in the Munc18-1-dependent SNARE zippering observed by us, we removed the NRD in the SNARE construct (*Figure 1B*) and repeated the above experiments. The NRD removal did not cause significant changes in the intrinsic stage-wise SNARE assembly (*Figure 1C,2A*, -NRD), which suggests that the NRD barely interacted with the SNARE motifs. Furthermore, the NRD removal did not alter the de novo SNARE assembly (marked by red arrows in *Figure 11A*, #1, and *9B*), indicating that the NRD is dispensable for this Munc18-1-dependent activity. However, Munc18-1 no longer stabilized the half-zippered state well (*Figure 11B,C* and *Figure 11-figure supplement 1*). Instead, Munc18-1 mainly stabilized a novel state (state 11) with a distinct extension that was 1–3.5 nm less

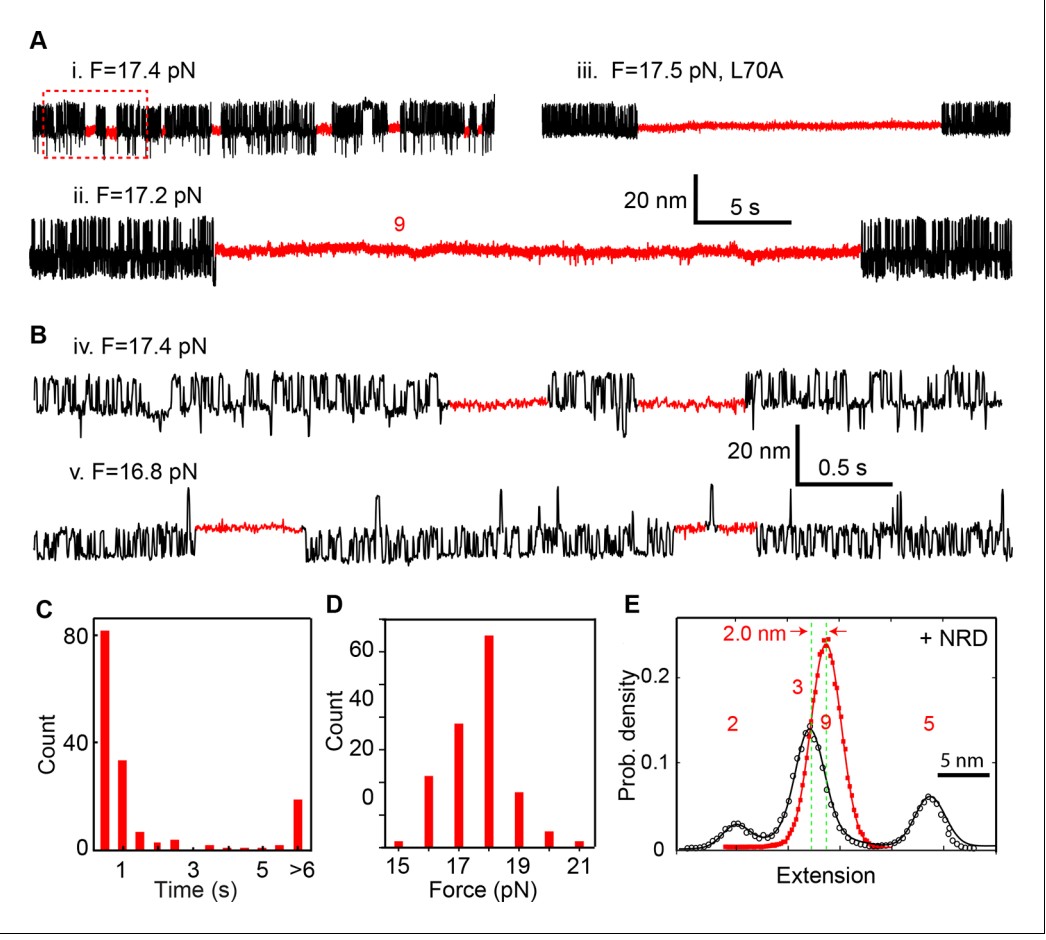

**Figure 10.** Munc18-1 stabilizes the half-zippered SNARE complex. (A–B) Extension-time trajectories of single wild-type or mutant (L70A) SNARE complexes under constant forces in the presence of 10 μM Munc18-1 in the solution. The red regions indicate the Munc18-1-bound state (state 9). A close-up view of the region in the dashed rectangle in i is shown in (B), iv. (C) Histogram distribution of the dwell times of the Munc18-1-stabilized half-zippered state. (D) Histogram distribution of the forces of the Munc18-1-stabilized half-zippered state. (E) Probability density distributions of the extensions of the Munc18-1-unbound states (black) and the Munc18-1-bound state (red) calculated from the corresponding regions in trace ii in A. Different peaks represent different SNARE folding states numbered as in *Figure 1D*.

than the NTD-unfolded state 5 (*Figure 11D* and *Figure 11-figure supplement 1*). The average extension difference between state 11 and state 5 is 2.5 (±0.7, N=16) nm, approximately equal to the extension shortening caused by binding of the Vc peptide (*Figure 3D*). This Munc18-1 activity occurred at a rate of 0.015 s⁻¹ measured from a total experimental time of 4018 s. Surprisingly, Munc18-1 now bound much more strongly to the SNARE complex. More than half the Munc18-1-bound states lasted more than 30 s. Therefore, the NRD removal significantly enhances the thermo-dynamic stability of Munc18-1 binding to the SNARE complex.

Munc18-1 also induced a minor state (state 10) that reversibly transited with the major state 11 (*Figure 11C,D*). State 10 had an average extension close to the half-zippered state 9, but with higher stability than state 9. When relaxing the Munc18-1-bound SNARE complex to a lower force, we saw an equilibrium shift from state 11 to state 10, reaching equilibrium at an average force of 14.0 (±1.2) pN with corresponding extension change of 5.7 (±0.6) nm (*Figure 11A*, #2, region marked by red rectangle). Munc18-1 was displaced from the SNARE complex in state 10 in a force range of 8-13 pN (indicated by the green arrow), with an average force of 10.3 pN. The displace-ment was accompanied by full SNARE assembly, which was confirmed by the FEC of the subsequent pulling phase. Thus, Munc18-1 attenuated SNARE NTD zippering in the absence of the NRD,

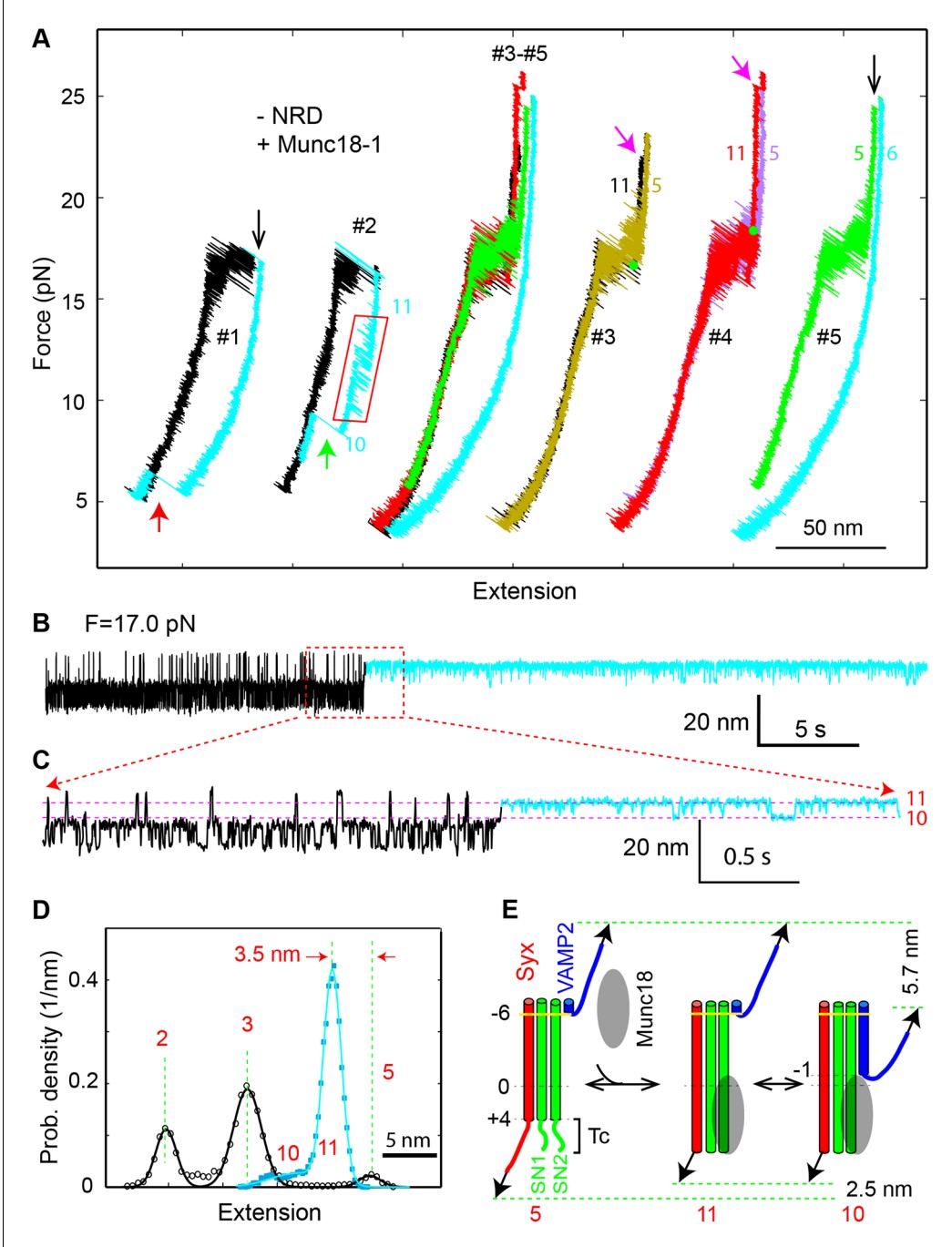

**Figure 11.** Munc18-1 stabilizes the half-zippered SNARE complex in an NRD-dependent manner and binds the t-SNARE complex to structure Tc in an NRD-independent manner. (**A**) FECs of the SNARE complex lacking the NRD in the presence of 10 μM Munc18-1 in the solution. The rectangle marks the Munc18-1-mediated reversible transition in FEC #2 between states 10 and 11 illustrated in E. The FECs #3-#5 were obtained from the same SNARE complex in three consecutive rounds of pulling and relaxation. The FECs of different rounds overlap well (#3-#5) but are shifted along the x-axis for clarity (#3, #4, and #5). The states corresponding to some regions of FECs are indicated by their corresponding state numbers (***Figures 11E and 1D***). In FECs #3 and #4 the point of Munc18-1 binding is marked by a green dot. (**B–C**) Extension-time trajectories of the SNARE complex lacking an NRD under constant forces in the presence of 10 μM Munc18-1. The cyan regions indicate the Munc18-1-bound states (states 10 and 11). A close-up view of the region in B in the dashed rectangle is shown in (**C**). (**D**) Probability density distributions of the extensions of the Munc18-1-unbound states (black) and the Munc18-1-bound states (cyan) calculated from the corresponding regions in the trace shown in B. Different peaks represent different

*Figure 11 continued on next page*

*Figure 11 continued*

SNARE folding states numbered in red as in E and *Figure 1D*. (E) Diagram illustrating effects of Munc18-1 on assembly of the SNARE complex lacking an NRD and the associated average extension changes of the different states. FECs, force-extension curves; NRD, N-terminal regulatory domain.

The following figure supplements are available for figure 11:

**Figure supplement 1.** (A–B) Extension-time trajectory of the SNARE complex lacking an NRD in the presence of 10 µM Munc18-1 in the solution.

**Figure supplement 2.** FECs of the SNARE complexes with the NRD (+NRD) obtained in the absence (−Munc18-1) or presence (+Munc18-1) of 10 µM Munc18-1.

**Figure supplement 3.** Extension-time trajectory of the wild-type SNARE complex with the NRD in the presence of 10 µM Munc18-1.

although Munc18-1 stabilized the NTD in state 9 in the presence of the NRD. Because state 10 could resist SNARE zippering at a lower external force than state 9, state 10 must have higher mechanical stability than state 9. The comparison suggests that Munc18-1 associated with the SNARE complex differently in the presence and absence of the NRD. Therefore, the NRD regulates Munc18-1's interaction with the SNARE bundle and is required to stabilize the SNARE complex in the half-zippered state (*Shen et al., 2007*; *Dulubova et al., 2007*).

## Munc18-1 induces t-SNARE folding

Next, we pulled the Munc18-bound state 11 to high forces. We saw a small extension jump in a force range of 22-28 pN, showing that the Munc18-bound state 11 unfolded into the NTD-unfolded state 5 (*Figure 11A*, indicated by magenta arrows). We inferred that state 11 has a conformation similar to the NTD-unfolded state 5, except that Munc18-1 is associated with a structured Tc in a way similar to the Vc peptide (*Figure 11E*). Indeed, upon SNARE binding Munc18-1 induced approximately equal extension shortening (2.5 ± 0.7 nm) from state 5 as the Vc peptide (2.3 ± 0.5 nm) did. Therefore, in the absence of NRD, Munc18-1 tightly bound t-SNARE and structured Tc. Because SNARE zippering was attenuated, Munc18-1 must also bind amino acids in t-SNARE N-terminal to −1 layer to interfere with NTD zippering (*Figure 11E*). Accordingly, we estimated that in state 10 VAMP2 was zippered to a position between −2 and −1 layers, in contrast with the half-zippered state 9 (between 0 and +1 layers). These observations suggest that Munc18-1 and the Vc peptide probably promote membrane fusion by the same mechanism: They structure Tc, stabilize the t-SNARE complex, and inhibit SNARE mis-assembly (*Pobbati et al., 2006*), which outcompete their adverse effects of attenuating SNARE zippering. Finally, we found that Munc18-1 occasionally (4 out of 113) bypassed the NRD to similarly attenuate zippering and structure Tc of the SNARE complex containing the NRD (*Figure 11- figure supplement 2,3*). This finding suggests that Munc18-1 may transit between the NRD-dependent and the NRD-independent SNARE-binding modes, consistent with previous observations (*Rathore et al., 2010*; *Hu et al., 2011*).

In conclusion, we observed that Munc18-1 interacts with the SNARE bundle in at least two modes and differentially regulates SNARE zippering: In an NRD-dependent mode, Munc18-1 stabilizes the half-zippered state, while in an NRD-independent mode, Munc18-1 structures Tc.

## Munc18-1 only loosely closes syntaxin

Upon relaxation of the completely unfolded SNARE complex in the presence of Munc18-1, partial refolding of the complex was sometimes observed (17 out of 113 SNARE complexes tested) (*Figure 12A*). The refolding process was reversible and fast in the force range of 4-8 pN, with an average equilibrium force of 6.1 (±0.8) pN and an extension change of ~4 nm (*Figure 12B*). This transition required both Munc18-1 and the NRD, because the transition disappeared when either Munc18-1 was omitted or the NRD was removed. We concluded that the transition resulted from the Munc18-1-mediated transition of the syntaxin molecule between the closed conformation and the open conformation (*Figure 12C*) (*Dulubova et al., 1999*). The structure of the open

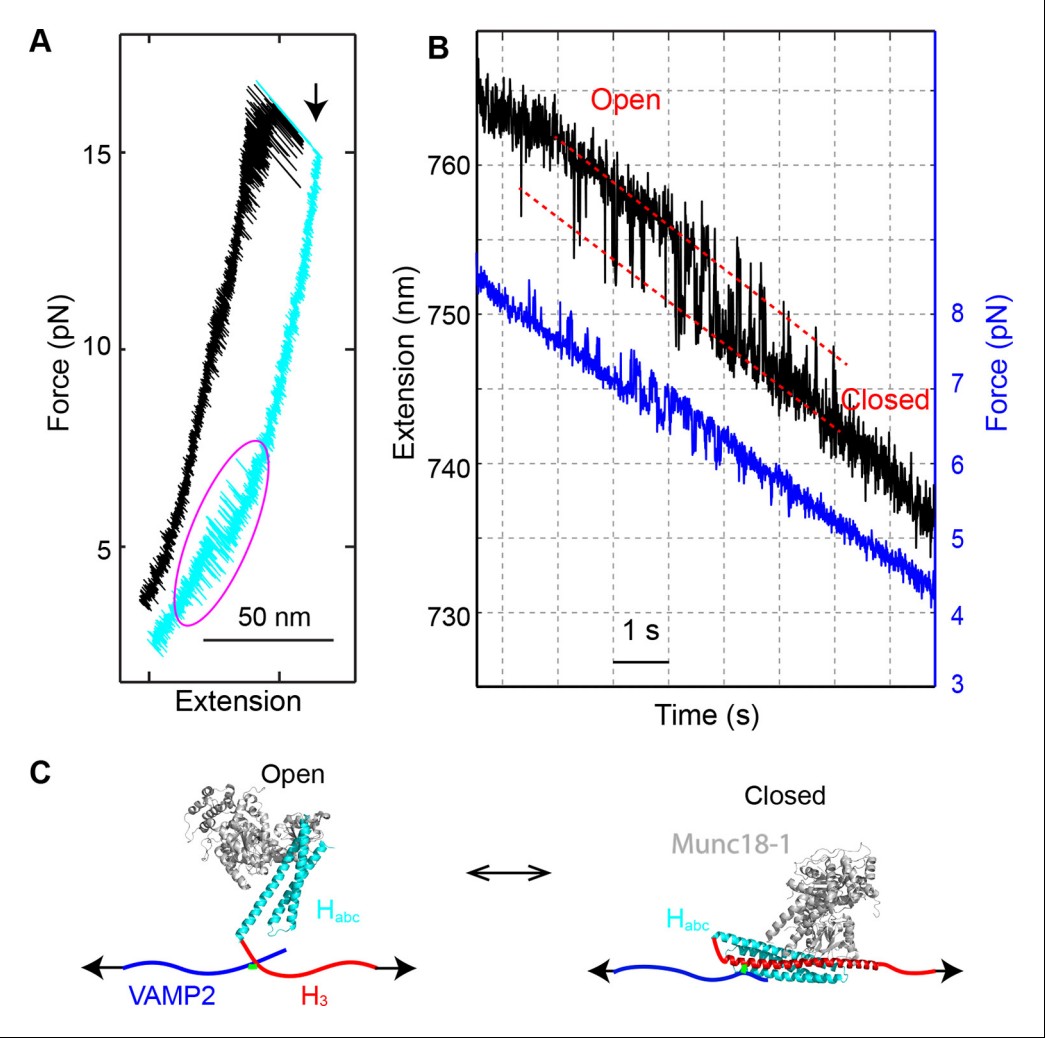

**Figure 12.** Munc18-1 only loosely closes syntaxin. (**A**) FECs obtained by pulling (black) and then relaxing (cyan) the SNARE complex. A magenta oval marks the Munc18-1-mediated syntaxin opening and closing transition illustrated in C. A black arrow indicates the t-SNARE unfolding event. (**B**) Time-dependent extension (black) and force (blue) showing opening and closing transitions of syntaxin as the SNARE complex was being relaxed in the presence of 10 µM Munc18-1. (**C**) Diagram of the derived syntaxin conformational transition.

conformation was not determined. Consequently, we estimated a lower limit of $\sim -7$ $k_BT$ for the free energy of the closed state relative to the open state, based on the measured mechanical work required to unfold the closed state. The relatively small energy difference revealed that the closed syntaxin could spontaneously open to participate in SNARE assembly. This finding is surprising, because it is generally believed that Munc18-1 tightly binds the closed syntaxin with nano-molar affinity, or $-20$ $k_BT$ association free energy (*Burkhardt et al., 2008*; *Misura et al., 2000*; *Colbert et al., 2013*). However, our measurement is consistent with a recent observation that syntaxin can spontaneously open (*Hu et al., 2011*; *Pertsinidis et al., 2013*). Given the overall low frequency to observe the syntaxin transition under our experimental conditions (<0.01 $s^{-1}$), Munc18-1 must remain attached to the syntaxin molecule during the whole syntaxin transition process (*Figure 12B*), suggesting a multivalent interaction between Munc18-1 and syntaxin (*Colbert et al., 2013*; *Hu et al., 2011*; *Burkhardt et al., 2008*). This finding shows that Munc18-1 binding and syntaxin closing do not necessarily coincide: syntaxin could stay in an open state while Munc18-1 remained attached to part of the syntaxin molecule, including the NRD (*Figure 12C*) (*Hu et al.,*

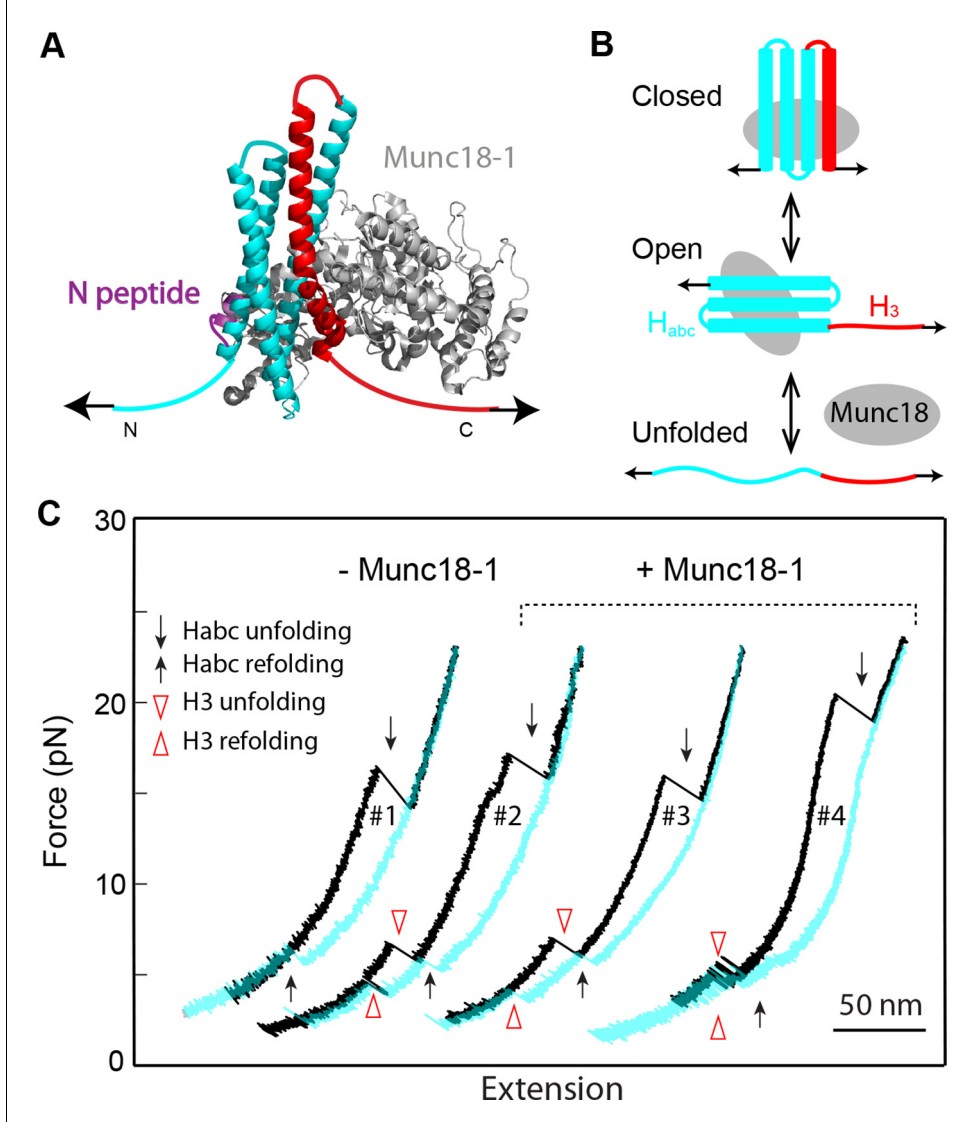

**Figure 13.** Opening the closed syntaxin by force. (**A**) Syntaxin construct pulled from its N- and C-termini to study the syntaxin conformational transition mediated by Munc18-1 (not shown). The syntaxin molecule contained a C-terminal Avi-tag and a cysteine inserted after amino acid D25, which was used to crosslink syntaxin to the DNA handle. (**B**) Predicted conformational transitions of the syntaxin molecule induced by force. (**C**) Representative FECs in the absence (−) and presence (+) of Munc18-1 confirming the predicted conformational transitions shown in (**B**). FECs, force-extension curves.

*2011*). The multivalent interaction also reconciles the marginal stability of the closed syntaxin observed by us and the overall high affinity between Munc18-1 and syntaxin.

To confirm the above observation, we pulled a single cytoplasmic syntaxin molecule from its C-terminus and a cysteine residue inserted after amino acid D25 following the N-peptide (*Figure 13A, B*). The N-peptide was free to bind Munc18-1 (*Burkhardt et al., 2008*; *Rathore et al., 2010*). Syntaxin alone unfolded in a force range of 15-25 pN and refolded at 3-7 pN, due to irreversible unfolding and refolding of the $H_{abc}$ domain (*Figure 13C*). In the presence of 10 µM Munc18-1 in the solution, we found another transition in the force range of 3-8 pN that only occurred in the presence of the folded $H_{abc}$, which represents the Munc18-1-mediated syntaxin opening and closing transition (*Figure 13B*). The transition was reversible (FECs #4 and #2), but with much smaller transition rate than that seen in the previous construct (*Figure 12B*). As a result, the extension flicking during

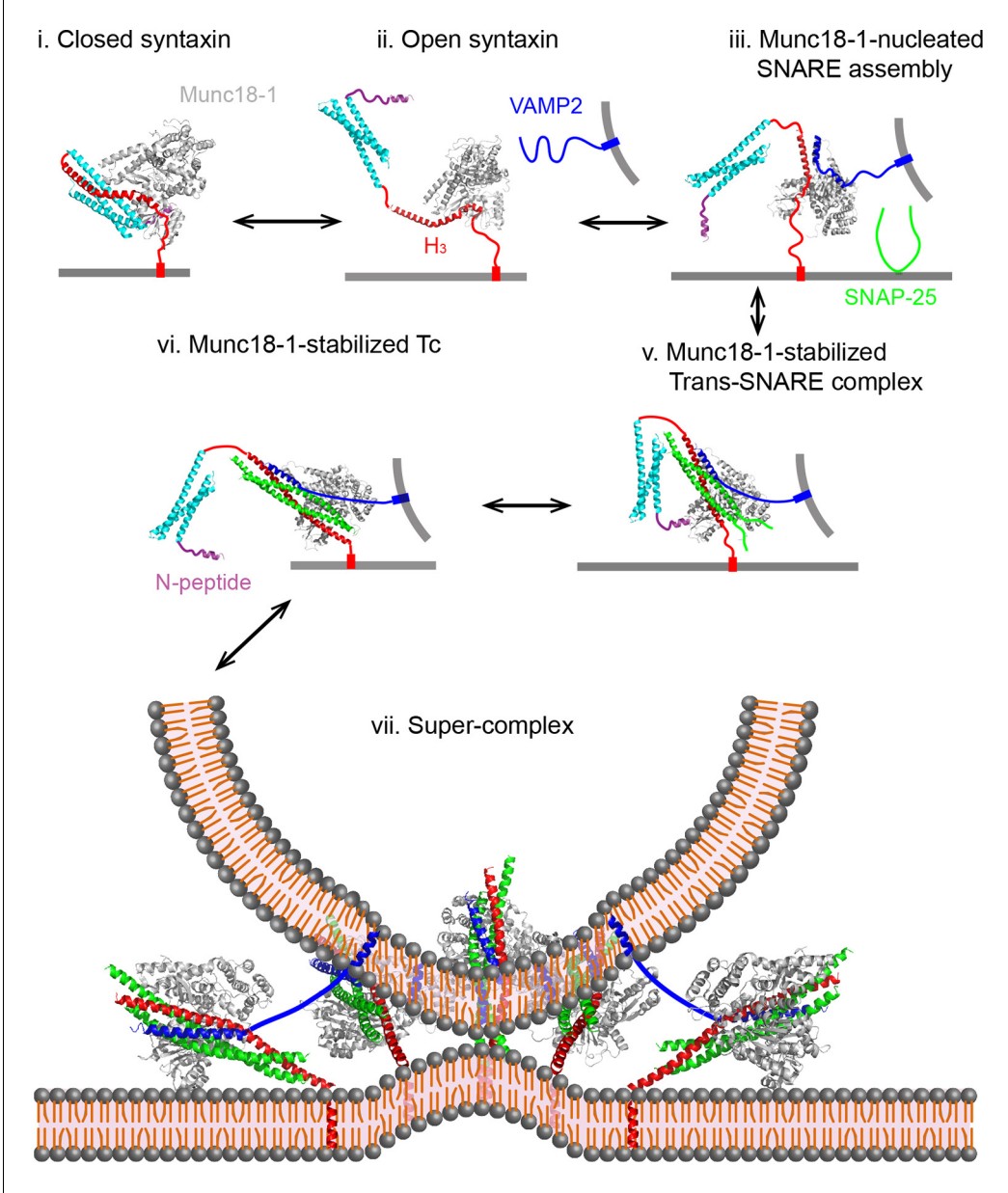

**Figure 14.** Working model of Munc18-1-regulated stage-wise SNARE assembly in membrane fusion. See the main text for details. Munc18-1 undergoes a conformation transition to open the syntaxin molecule (from states i to ii) and form the VAMP2 binding surface (*Hu et al., 2011*; *Parisotto et al., 2014*). Thus, syntaxin opening and the template formation in state iii may be cooperative, which likely leads to a transient open syntaxin conformation shown in state ii.

pulling or relaxation was not seen in some FECs, such as FEC #3 in *Figure 13C*. Thus, the syntaxin transition took place in a similar low-force range with a similar extension change as observed in the previous SNARE construct (*Figure 12C*), confirming the marginal stability of the closed syntaxin state. As a result, the presence of NRD did not significantly reduce the probability for Munc18-1 to mediate de novo SNARE assembly (*Figure 9B*), although the unfolded SNARE motif in syntaxin could alternatively enter the closed state (*Figure 12C*). The different transition kinetics seen in the two syntaxin constructs is not surprising, because the syntaxin molecule was pulled in approximately perpendicular directions (compare *Figure 13A* to *Figure 12C*). It is known that protein folding kinetics, but not folding energy, strongly depends on the pulling direction (*Gao et al., 2011*).

### Energetics and kinetics of SNARE assembly underlying exocytosis

Our above measurements provide a basis to understand SNARE-mediated membrane fusion. We found that SNARE assembly exhibited different sensitivity to mutations that correlates with their various phenotypes in vivo. Compared to the wild-type SNARE complex, the NTD mutation V39A/V42A reduces the number of docked vesicles, but does not change the $Ca^{2+}$-triggered fusion rate of the vesicles in the readily releasable pool in chromaffin cells, as was derived from electrophysiological measurements (*Walter et al., 2010*). In parallel, our measurements showed that the same mutation destabilized NTD by 3 $k_BT$ with no change in CTD zippering energy (*Figure 7*). The comparison suggests that NTD association is responsible for docking, but not vesicle fusion and that the NTD folding energy dictates the number of docked vesicles. Accordingly, the NTD association and dissociation reaches dynamic equilibrium, consistent with dynamic vesicle docking and undocking observed in vivo and in vitro (*Zenisek et al., 2000*; *Zhang et al., 2015*). Thus, we predicted that the reduced NTD association rate of the mutant observed by us (*Figure 5A*) should decrease the fusion rate of the vesicles in the sustained releasable pool (SRP). Indeed, the SRP fusion rate of the mutant is reduced by ~50% (*Walter et al., 2010*), confirming the prediction. Therefore, NTD association energy and kinetics contribute to the magnitude and rate of vesicle docking, respectively.

Mutations that decreased CTD zippering energy by ~10 $k_BT$, including L70A (+4) and F77A (+6) in VAMP2 and M71A/I192A (+5) in SNAP-25B, also dramatically reduce the rate of exocytosis (*Walter et al., 2010*; *Mohrmann et al., 2010*). Interestingly, I67N (+4) in SNAP-25B is a dominant mutation identified from a patient with myasthenia, ataxia, and intellectual disability (*Shen et al., 2014*). It is found that this mutation impaired neuromuscular transmission. These results demonstrated that the layers +4, +5, and +6 are crucial for CTD zippering and membrane fusion. In contrast, syntaxin T251I (+7) in syntaxin enhanced CTD zippering and is found to increase both spontaneous and evoked exocytosis (*Lagow et al., 2007*). However, flies carrying this mutation become paralyzed at 38°C. These findings are consistent with the existence of a metastable partially zippered SNARE complex in the docked state (*Gao et al., 2012*; *Kyoung et al., 2011*; *Lou et al., 2015*; *Shin et al., 2014*) and mutations that stabilize CTD destabilize the half-zippered SNARE complex, leading to mis-regulated exocytosis and reduced precision of neurotransmission. Thus, the stabilities of different SNARE domains appear to be optimized to balance the accuracy and speed of membrane fusion. Finally, other CTD mutations, A67S (+3) and L84A (+8), do not significantly change the CTD folding energy and rate, nor exocytosis (*Walter et al., 2010*). In conclusion, CTD zippering directly drives membrane fusion by tightly coupling its folding energy to lowering the energy barrier of membrane fusion, thereby increasing the fusion rate.

R56A reduced SNARE zippering energy (*Figure 7*) and was found to impair membrane fusion in some reports (*Ossig et al., 2000*). Other MD mutations L60A (+1) and L63A (+2) abolished the independent MD transition and thus changed the SNARE folding pathway. Yu et al. recently examined the impact of both mutations in synaptic exocytosis using cultured mouse neurons (*Yu et al., 2015*). They found that the two mutations abolished both spontaneous and evoked exocytosis, implying an important role of independent MD assembly in exocytosis. The comparison suggests that an obligate pathway of SNARE assembly may be required for membrane fusion.

## Discussion

Synaptic SNARE assembly underlies the strength, speed, and precision of neurotransmission, as well as many neurological disorders. Because SNAREs assemble in multiple stages, SNARE folding intermediates, energetics, and kinetics are intrinsically coupled, making it difficult to characterize SNARE assembly using traditional experimental approaches. By applying high-resolution optical tweezers, we have systematically measured the folding energies and kinetics of both wild-type and mutant SNARE complexes in the absence and presence of Munc18-1. Because of the stage-wise SNARE assembly, crosslinking at the N-terminal edge of NTD (*Figure 1B*) slightly affects the measured NTD energy, but not energies and kinetics of other domains (*Gao et al., 2012*; *Zorman et al., 2014*). In addition, the NTD energies of the mutants relative to that of the WT barely change with the crosslinking site. Combining our findings with previous results on exocytosis, we confirmed distinct functions of different SNARE assembly stages in synaptic exocytosis.

We showed that tight thermodynamic coupling requires specific SNARE assembly intermediates and pathways. MD mutations cause significant heterogeneity in SNARE assembly kinetics and/or

abolish independent MD assembly. Furthermore, the MD is targeted by many regulatory proteins that control SNARE assembly, including complexin (*Chen et al., 2002*; *Sudhof and Rothman, 2009*; *Krishnakumar et al., 2011*) and synaptotagmin (*Choi et al., 2010*; *Zhou et al., 2015*; *Chapman, 2008*). Thus, the SNARE engine consists of four stages of zippering in NTD, MD, CTD, and LD with distinct functions. The NTD is responsible for initiating specific formation of the trans-SNARE complex and docking vesicles to the plasma membrane (*Walter et al., 2010*). The MD serves as a checkpoint to further ensure proper SNARE assembly and as a master switch to control CTD assembly. The CTD acts as a power stroke that directly drives membrane fusion. Continued zippering to LD and the transmembrane domains may provide additional energy for membrane fusion (*Stein et al., 2009*) or alter the interactions between SNAREs and membranes to facilitate fusion (*Ngatchou et al., 2010*; *Liang et al., 2013*; *Honigmann et al., 2013*).

We found that Munc18-1 regulates every stage of SNARE assembly. Combining with previous results, we propose a working model to account for the multiple roles of Munc18-1 in stage-wise SNARE zippering and membrane fusion (*Figure 14*). First, we confirmed the previous observation that Munc18-1 only loosely closes syntaxin and associates with open syntaxin (states i and ii) (*Hu et al., 2011*; *Christie et al., 2012*). The marginal stability of the closed syntaxin may explain why Munc13 further enhances syntaxin opening, despite the weak interaction between Munc13 and syntaxin (*Ma et al., 2013*; *Ma et al., 2011*). Second, we observed that Munc18-1 efficiently initiates SNARE assembly (from state ii to state v). Because this activity is NRD-independent, Munc18-1 likely directly binds SNARE motifs to promote SNARE assembly (states ii and iii) (*Shi et al., 2011*). Third, Munc18-1 stabilizes the half-zippered SNARE complex in an NRD-dependent manner (state v). This activity is strongly force-dependent and disappears at a force below ~15 pN. Fourth, we found that Munc18-1 associates with the t-SNARE complex (*Zhang et al., 2015*; *Pertsinidis et al., 2013*) and induces Tc folding in an NRD-independent manner (state vi). Finally, we did not find evidence that Munc18-1 stabilized the fully assembly SNARE complex in our single-molecule experiments, consistent with a recent report (*Zhang et al., 2015*). Although Munc18-1 is shown to stabilize trans-SNARE complexes in another report, the effect of stabilization is only minor (*Lou et al., 2015*). Thus, we conclude that Munc18-1 mainly enhances membrane fusion by catalyzing SNARE assembly, rather than by stabilizing the fully assembled SNARE complex. Consistent with this view, the rate of membrane fusion poorly correlate with the binding affinity between SM proteins and the fully assembled SNARE complexes (*Shen et al., 2007*; *Shen et al., 2015*). In particular, SNARE mutations that compromise interactions between the fully assembled SNARE complexes and their cognate SM proteins can support membrane fusion (*Peng and Gallwitz, 2004*). Taking together, we conclude that Munc18-1 mainly acts on the folding intermediates of SNAREs to chaperone their assembly.

The Munc18-1-regulated SNARE assembly observed by us corroborates or clarifies many key findings on the role of SM proteins in SNARE assembly and membrane fusion. During revision of this work, Baker et al. reported the crystal structures of fungus SM protein Vps33 bound with its cognate individual Vam3 and Nyv1 SNAREs (*Baker et al., 2015*). These proteins mediate vesicle fusion with vacuoles or endosomes. In particular, Vam3 and Nyv1 are homologs of syntaxin and VAMP2 belonging to the same Qa- and R-SNARE families (*Fasshauer et al., 1998*), respectively. They found that Vps33 interacts with the entire Qa- and R-SNARE motifs through two non-overlapping surfaces on Vps33 and aligns the SNARE helical NTDs in proximity and register (state iii). Because the structures of all SNARE complexes and SM proteins are highly conserved, they predicted that SM proteins serve as a template to initiate SNARE assembly into a partially zippered complex. Our discoveries of Munc18-1-mediated de novo SNARE assembly and half-zippered SNARE complex strongly support this prediction, which in turn offers a molecular mechanism for our observations (state iii). In addition, we previously showed that representative SNARE complexes mediating diverse membrane trafficking pathways slowly initiate their NTD assembly and pause in half-zippered states. These observations make SM-mediated SNARE assembly necessary. Our findings also corroborates that Munc18-1 enhances assembly of trans-SNARE complexes, thereby promoting liposome-liposome fusion (*Rathore et al., 2010*; *Shen et al., 2007*; *Shen et al., 2015*). Finally, the N-peptide is essential for recruiting Munc18-1 to the fusion site to assist SNARE assembly (*Rathore et al., 2010*). Interestingly, once the trans-SNARE complexes are formed, the N-peptide is not required and can be cleaved from syntaxin SNARE motif without compromising the subsequent liposome fusion in vitro, indicating that Munc18-1 binds the trans-SNARE complex in an NRD-independent manner before fusion (state vi). Consistent with this observation, we considered that after trans-SNARE assembly,

Munc18-1 switches to the NRD-independent Tc binding mode observed by us. Munc18-1 then induces Tc folding and greatly stabilizes the t-SNARE to facilitate SNARE assembly and membrane fusion similar to Vc and Vn peptides (*Melia et al., 2002*; *Pobbati et al., 2006*; *Li et al., 2014*). Moreover, our results are consistent with other potentially essential roles of SM proteins in membrane fusion, which are to protect disassembly of t- and trans-SNARE complexes by NSF (*Xu et al., 2010a*; *Lobingier et al., 2014*; *Zhao et al., 2015*) and to proofread SNARE zippering (*Lobingier et al., 2014*). These comparisons suggest that SM proteins act as chaperones to enhance SNARE assembly, which is essential for membrane fusion.

Our findings contrast with previous reports that Munc18-1 reduces the rate of SNARE assembly in an NRD-dependent manner, probably due to different experimental conditions (*Burkhardt et al., 2008*). We noticed that Munc18-1 and other SM proteins significantly enhance membrane fusion only under special conditions. Shen and co-workers showed that incubating SNARE-anchored liposomes in the presence of Munc18-1 at 4°C or adding crowding agents, such as 100 mg/ml Ficoll 70, in the fusion reaction is required to observe Munc18-1-enhanced liposome fusion (*Shen et al., 2007*; *Yu et al., 2015*). In general, SM proteins appear to weakly associate with both Qa- and R-SNAREs to form the template that initiates SNARE assembly (*Parisotto et al., 2014*; *Xu et al., 2010b*; *Baker et al., 2015*; *Lobingier and Merz, 2012*). The energy barrier for syntaxin to transit from a closed conformation to an open conformation further impedes formation of the template between Munc18-1 and syntaxin (*Hu et al., 2011*; *Baker et al., 2015*). The use of N-terminally crosslinked SNARE constructs in our experiments may facilitate Munc18-1 binding to both syntaxin and VAMP2 to form the template required for SNARE assembly (state iii). In addition, the presence of a single SNAP-25 molecule in our experiments may increase the contrast to observe Munc18-1-mediated de novo SNARE assembly. With a high concentration of SNAP-25 in the solution in ensemble experiments, the SM-mediated SNARE assembly may be obscured by the high rate of intrinsic SNARE assembly (*Baker et al., 2015*).

Evidence suggests that Munc18-1 greatly accelerates assembly of the SNAREs on the liposomes that are already docked (*Shen et al., 2007*; *Yu et al., 2015*). Munc18-1 may orchestrate a specific number of trans-SNARE complexes (state vii) (*Mohrmann et al., 2010*; *Karatekin et al., 2010*; *Megighian et al., 2013*; *Hernandez et al., 2014*; *Lu et al., 2008*) to cooperatively zipper. Our measurements showed that most of the mutant SNARE complexes are still more stable than many SNARE complexes mediating other membrane trafficking pathways (*Zorman et al., 2014*). As a result, SNARE-mediated fusion in the absence of Munc18-1 is remarkably insensitive to CTD folding energies (*Zorman et al., 2014*; *Shen et al., 2007*). We hypothesize that Munc18-1 helps to form a super-complex containing Munc18-1, trans-SNARE complexes, and other regulatory proteins with approximately fixed stoichiometry to efficiently and specifically drive membrane fusion. Without Munc18-1-dependent constraints on the number of trans-SNARE complexes, the energy loss caused by SNARE mutation may be compensated by the energy gain from recruitment of more SNARE complexes to the fusion site. As a result, membrane fusion mediated by SNARE alone has variable number of SNARE complexes involved (*Hernandez et al., 2014*), which causes non-specific membrane fusion. Thus, Munc18-1 intimately chaperone SNARE folding from syntaxin to multiple trans-SNARE complexes, making it indispensable for SNARE-mediated membrane fusion.

## Materials and methods

### Protein purification, labeling, and SNARE complex formation

All recombinant proteins were expressed in BL21 *E. coli* cells and purified using His-tags. Syntaxin was biotinylated at the Avi-tag in vitro using the biotin ligase (*Gao et al., 2012*). The His-tag was removed from purified syntaxin and VAMP2, but kept in other proteins. SNARE complexes were formed overnight by mixing syntaxin and VAMP2 in 1:1 molar ratio and an excessive amount of SNAP-25B and then purified using the His-tag on SNAP-25.

### Dual-trap optical tweezers

The dual-trap optical tweezers were home-built and calibrated as previously described (*Gao et al., 2012*; *Sirinakis et al., 2012*; *Moffitt et al., 2006*). Briefly, a laser beam with a wavelength of 1064 nm was expanded, collimated, and split into two orthogonally polarized beams. One beam was

reflected by a mirror attached to a nano-positioning stage that could tip/tilt in two axes with high resolution (Mad City Labs, WI). The two beams were then combined, further expanded, and focused by a water-immersion 60X objective with a numerical aperture of 1.2 (Olympus, PA) to form two optical traps. One optical trap was moved by turning the mirror to control the force applied to a single molecule. The outgoing laser beams were collimated by an identical objective, split again by polarization, and projected to two position-sensitive detectors (Pacific Silicon Sensor, CA) to detect bead positions using back-focal-plane interferometry. The force constants of both optical traps were calibrated by Brownian motion of the trapped beads (*Zhang et al., 2012*). The force, extension, trap separation, and other experimental parameters were acquired at 20 kHz, filtered online to 10 kHz, and stored on hard-disk.

## Single-molecule experiments

The SNARE complex was mixed with the DNA handle at a molar ratio of 50:1 and crosslinked by exposing the solution to air overnight to oxidize the thiol groups (*Gao et al., 2012*). An aliquot of the mixture was bound to anti-digoxigenin antibody-coated polystyrene beads. A single such bead was subsequently captured in one optical trap and brought close to a streptavidin-coated bead in a second trap to form the protein-DNA tether between the two beads. All SNARE pulling experiments were carried out in phosphate-buffered saline supplemented with an oxygen-scavenging system at room temperature.

## Data analysis

The data analysis yielding the intermediates and energies was performed as previously described (*Gao et al., 2012*) and detailed in a manuscript using the SNARE complex containing the VAMP2 R56A mutant as an example (*Rebane, et al., 2016*). We normally performed the HMM analysis for the whole extension-time trajectory obtained at each trap separation that typically lasted 5–300 s, after the trajectory was mean-filtered to 5 kHz or 1 kHz. We calculated the histogram distribution of the extension and determined the number of states by fitting the distribution with multiple Gaussian functions. Then, we optimized the parameters in the hidden Markov model using gradient descent (*Qin et al., 2000*). Finally, the idealized extension trajectories were calculated using the Viterbi algorithm (*Rabiner, 1989*).

We chose the contour length of the unfolded and stretched polypeptide $l$ as a reaction coordinate to describe SNARE folding along a pathway inferred from the crystal structure of the fully assembled SNARE complex (*Sutton et al., 1998*; *Stein et al., 2009*). We first derived the conformation of the t-SNARE complex in state 5 based on its extension relative to the fully unfolded state 6 (*Gao et al., 2012*) (*Figure 1D*) and the Vc-bound state 8 (*Figure 3*). The derived t-SNARE conformation is also supported by our experiments directly pulling a single t-SNARE complex from the C-termini of syntaxin and SNAP-25 (unpublished results). We assumed that NTD and MD zippering proceeded by folding of VAMP2 on the structured t-SNARE template and that CTD zippering was accompanied by concurrent folding of both VAMP2 and Tc. The LD transition was considered to be symmetrical zippering of syntaxin and VAMP2 and was characterized as before (*Gao et al., 2012*). To derive conformations of the intermediate and transition states in the folding pathway from the overlapping CTD, MD, and NTD transitions, we defined a simplified energy landscape $(l_i, V_i)$, characterized by the contour length $l_i$ of $i$-th intermediate state or transition state and its associated free energy $V_i$ at zero force. These parameters were determined by fitting the model-based calculations to the experimental measurements. To do so, we first established the experimental observables as a function of the simplified energy landscape. We calculated the total extension $X_i$ and energy $G_i$ for each state in the presence force. Both quantities constituted contributions from three components: the unfolded polypeptide, the folded portion of the protein, and the DNA handle (*Figure 2—figure supplement 3*). The total energy additionally contained the potential energy of two beads in optical traps. Thus, the extension of the protein-DNA tether was calculated as

$$X_i = x_i^{(m)} + H_i + x_i^{(DNA)} \tag{1}$$

Here, the force-dependent extension of the unfolded polypeptide or the DNA handle is described by the Marko-Siggia formula for a semi-flexible worm-like chain (*Marko and Siggia, 1995*), i.e.,

$$F_i = \frac{k_B T}{P}\left[\frac{1}{4\left(1-\frac{x_i}{L}\right)^2} + \frac{x_i}{L} - \frac{1}{4}\right], \tag{2}$$

where $F_i$ is the average force of state $i$ and $P$ and $L$ are the persistence length and the contour length of the chain, respectively. We chose $P$ = 40 nm and $L$ = 768.4 nm for DNA (*Bustamante et al., 1994*) and $P$ = 0.6 nm and $L = l_i$ for the polypeptide (*Gao et al., 2011*; *Xi et al., 2012*). The extension of the folded protein portion was calculated based on a freely jointed chain model (*Smith et al., 1992*)

$$H_i = -\frac{k_B T}{F_i} + h_i \coth\left(\frac{F_i h_i}{k_B T}\right), \tag{3}$$

where $h_i$ is the size the folded protein portion in state $i$ along the pulling direction, which was measured from the crystal structure as a function of the contour length $l$ of the unfolded polypeptide. The tether extension is related to the experimental control parameter trap separation $D$ by the following formula

$$D = X_i + \frac{F_i}{k_{traps}} + R, \tag{4}$$

where $F_i/k_{traps}$ is the total bead displacement with $k_{traps} = k_1 k_2/(k_1+k_2)$ the effective trap stiffness of the two optical traps and $R$ the sum of the bead radii. Given the experimental conditions and contour length of a state $l_i$, we could solve its state force $F_i$ by substituting *Equations 1–3* into *Equation 4*. Similarly, we could calculate the total energy of the system as

$$G_i = G_i^m + V_i + G_i^{(DNA)} + \frac{F_i^2}{2k_{traps}}, \tag{5}$$

where $G_i^{(m)}$, $G_i^{(DNA)}$, and $F_i^2/(2k_{traps})$ are the energies of the unfolded polypeptide, the DNA handle, and the beads in optical traps, respectively. The former two energies result from entropy changes of the worm-like chains due to stretching, which were calculated as

$$G_i^{(m \text{ or } DNA)} = \frac{k_B T}{P}\frac{L}{4\left(1-\frac{x_i}{L}\right)}\left[3\left(\frac{x_i}{L}\right)^2 - 2\left(\frac{x_i}{L}\right)^3\right]. \tag{6}$$

Given the state energies, we computed the state populations based on the Boltzmann distribution and the transition rates in terms of Kramers' formula. Finally, we simultaneously fit the calculated state forces, populations, transition rates, and extension changes to the corresponding measurements using the nonlinear least-squares method, yielding the best-fit parameters for the simplified energy landscape.

## Acknowledgements

We thank James Rothman, Frederick Hughson, Thomas Söllner, Jingshi Shen, Erdem Karatekin, Shyam Krishnakumar, Frederic Pincet, Xinming Zhang, and Junyi Jiao for discussion and reading the manuscript, Hong Qu, Bowei Su, and Haijia Yu for help in experiments, and Mengze He for preparing the videos. This work was supported by the NIH grant RO1GM093341 to Y. Z. Research reported in this publication was also supported in part by the Brain Research Foundation and by the Raymond and Beverly Sackler Institute for Biological, Physical and Engineering Sciences at Yale.

## Additional information

### Funding

| Funder | Grant reference number | Author |
| --- | --- | --- |
| National Institutes of Health | GM093341 | Yongli Zhang |

Raymond and Beverly Sackler Institute for Biological, Physical and Engineering Sciences, Yale University

Yongli Zhang

The funders had no role in study design, data collection and interpretation, or the decision to submit the work for publication.

## Author contributions

LM, GY, Conception and design, Acquisition of data, Analysis and interpretation of data, Drafting or revising the article, Contributed unpublished essential data or reagents; AAR, ZX, YK, YG, Conception and design, Acquisition of data, Analysis and interpretation of data, Drafting or revising the article; YZ, Mentoring, Conception and design, Analysis and interpretation of data, Drafting or revising the article

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
