## [Decision Letter]

Thank you for submitting your work entitled "Munc18-1-regulated stage-wise SNARE assembly underlying synaptic exocytosis" for peer review at *eLife*. Your submission has been favorably evaluated by Vivek Malhotra (Senior Editor), and three reviewers, one of whom is a member of our Board of Reviewing Editors.

The reviewers have discussed the reviews with one another and the Reviewing Editor has drafted this decision to help you prepare a revised submission.

Summary:

This paper describes a technically impressive study, in which the authors used an optical tweezers to map out the energetics and kinetics of neuronal SNARE assembly. Two technical features distinguish this work from earlier efforts. First, the authors figured out how to crosslink VAMP and syntaxin so that they could observe assembly as well as disassembly. Second, they determined assembly energetics/kinetics for a well-chosen set of VAMP (and a small number of syntaxin and SNAP-25) mutants. They then went on to determine assembly energetics/kinetics in the presence of Munc18. The authors demonstrate: (1) that regions within the SNARE bundle are responsible for a series of assembly intermediates (at least under the conditions of the experiment) that they propose correspond to well-characterized stages in vesicle docking and membrane fusion: (2) that Munc18 can bind and influence the structures and energetics of SNARE assembly intermediates, providing support for the idea that Munc18 functions as an assembly chaperone.

Essential revisions:

The paper is, in some respects, two separate reports. In one, the authors characterize 15 different SNARE-complex mutants, previously studied by different methods and in different experimental setting. As the authors correctly point out, a coherent treatment by a uniform method is needed. In the other, the authors examine the effects of Munc18 on assembly and disassembly of single SNARE complexes. There are many new observations and insights, but the conclusions are less convincing than those of the first part (see below).

Major comments:

Overall:

(a) The paper’s structure needs to be reorganized. Some important points are treated summarily (see last paragraph under "part 1", below), while some irrelevant ones are emphasized.

(b) Please take care not to base structural interpretations on single distance measurements. While the models for states 1-8 are undeniably useful, and while the mutagenesis results as well as previous studies provide some additional support, the models should be presented and discussed more cautiously. Even the presence or absence of, for example, SNAP-25 in each of these states does not strike the reviewers as a foregone conclusion.

(c) As a specific example of interpretive ambiguity, the authors assume for the unfolding state 4 in Figure 1 that syntaxin (and probably also, SNAP-25) is partially disassembled up to layer +5. It is also possible, however, that the t-SNARE α-helices remain intact, while unfolding of VAMP2 progresses beyond the point shown, for example, up to layer -1 or -2. How do the authors differentiate between these possibilities? Since the C-terminal fragment of VAMP2 (VAMP2 49-96) induces tight folding of the C-terminal part of the t-SNARE proteins (many data in the previous literature and Figure 5 in this manuscript), addition of the C-terminal VAMP2 fragment should decrease the extension value of states 4 and 5 if the t-SNARE complex is indeed partially melted as the authors suggest. The authors might also (in part 2) repeat this experiment with Munc18, because they claim that Munc18, like the C-terminal VAMP2 fragment, chaperones the C-terminal end of the t-SNARE complex.

(d) The suggested switching of Munc-18 interactions likewise appears to be arbitrary. The authors make no effort to justify that interpretation in terms of what is known about Munc18 structure and biochemistry. There are many ways in which two apparently distinct activities can be interpreted, without requiring distinct modes of interaction. The authors overstate the case by saying (rather pompously) "we discovered (sic) that Munc18-1 interacts with the SNARE bundle in at least two distinct modes".

(e) The authors claim that they have "quantitatively verified the tight thermodynamic coupling between SNARE zippering and membrane fusion and revealed distinct functions of different SNARE assembly stages." They have done nothing of the kind. They have done a fine job of detecting intermediates in a useful and valuable model, in which individual SNARE complexes are pulled and relaxed, and they can (and should) be pleased with that accomplishment. But when several SNARE complexes are pulling on the same two membranes and the membranes are pulling back on them, as is the case for exocytic fusion (indeed, membrane fusion generally), then the relative heights of barriers, etc., might well be different. That difference doesn't diminish the value of the numbers provided here, but it does rule out egregious overstatement. Even the interpretation in the third paragraph, subsection “Energetics and kinetics of SNARE assembly underlying exocytosis”, seems stretched – why do the phenotypes of those mutations reveal "an essential role of independent MD assembly"? Yes, messing up the MD messes up the mouse, but how does even the correspondence to the kinetic observations reported here demonstrate "independent" (rather than "concerted", with the other segments) assembly?

Part 1:

Although the authors basically performed the same experiments as in Gao et al. (2012), there seem to be two differences. The first is that unfolding in the C-terminal half of the SNARE complex, previously treated as one process, has now been resolved into CTD and MD unfolding (states 3 and 4), presumably because they did less temporal averaging of the data and plotted the extension traces over 1kHz. The second difference is that they moved the crosslinking site to layer -6. This difference likely separated NTD unfolding (state 5) from dissociation of SNAP-25 (state 6) more clearly than in Gao et al. (2012), in which the crosslinker was at layer -7. Thus, before dissociation of SNAP-25 at a higher force, they could see reversible NTD unfolding and refolding transitions.

In Figure 2 and Figure 3, the authors used the high temporal (over 1 kHz) and spatial resolution provided by the optical tweezers to resolve detailed structural transitions occurring in a single neuronal SNARE complex – a substantial technical accomplishment. They resolve the C-terminal unfolding into two intermediates (3 and 4). The transition between the two involves unfolding (or refolding) of the middle part between the +2 and the zeroth layers (but this analysis may need to be refined). The mutations in the +1 or +2 layer accordingly affect the stability of these intermediates. Thus, the probability of observing intermediate 3 is selectively reduced (i.e., the unfolding process pauses less frequently at the +2 layer).

The authors claim that "the middle domain plays a pivotal role in robust and correct stage-wise SNARE assembly, and its mutation leads to changes in SNARE assembly pathway and accuracy" and that "the MD serves as a checkpoint to further ensure proper SNARE assembly as a master switch to control CTD assembly". It is hard to understand how the reduced observation of the intermediate 3 (observed for the L60A and L63A mutants) supports these rather strong arguments. In the third paragraph of the subsection “Energetics and kinetics of SNARE assembly underlying exocytosis”, the authors allude to unpublished results, but this citation is not an appropriate way to justify the arguments. The authors should either weaken these claims or provide experimental evidence to support their arguments.

Part 2:

Data showing different effects of Munc18 are mixed confusingly together, making part 2 extremely hard to follow, even for a reader who knows the field. In particular, in Figure 5 and Figure 6, rearrangements of the figure subpanels would improve readability. For example, the subpanels, Figure 5 and Figure 6 report that Munc18 helps reassembly of the SNARE complex after complete unfolding of the t-SNARE pre-complexes, so that these panels can be grouped together to make one figure. On the other hand, the subpanels including Figure 5 to G and Figure 6 to D report stabilization of the half-zipped SNARE complex by Munc18, which may be grouped together to make another figure.

From the extension difference between states 8 and 5 (Figure 6), observed only in the absence of NRD (N-terminal Regulatory Domains), the authors argue that Munc18 would bind to the C-terminal part of the t-SNARE proteins and induce a coil-to-helix transition. But then, state 4 with bound Munc18 should have a smaller extension than state 4 without Munc18.

Figure 7 and the accompanying discussion are not very clear, and the reviewers found the conclusions unconvincing. For example, the opening and closing of syntaxin induced by Munc 18 (Figure 7) is intriguing, but then why do authors not observe reversible transitions between open and closed syntaxin in the more direct pulling study shown in Figure 7?

Overall recommendation:

1) Present the structural models for the kinetic states more cautiously and explain any alternatives.

2) Support the interpretation of the role of the MD with data (and pay attention to other comments under "part 1", above).

3) When revising what we have called "part 2", comply with each of the three paragraphs under "part 2", above.

4) Provide details (in the Methods) for the nonlinear least-square fitting that converts dwell times to deltaG (or deltaG double dagger).

[Editors' note: further revisions were requested prior to acceptance, as described below.]

Thank you for submitting your work entitled "Munc18-1-regulated stage-wise SNARE assembly underlying synaptic exocytosis" for consideration by *eLife*. Your article has been reviewed by two peer reviewers, and the evaluation has been overseen by a Reviewing Editor and Vivek Malhotra as the Senior Editor.

The reviewers have discussed the reviews with one another and the Reviewing Editor has drafted this decision to help you prepare a revised submission.

Summary:

The reviewers agreed that the manuscript is greatly improved, but it remains, despite the revision, challenging and complex. In particular, two parts need attention, as follows.

Essential revisions:

A) The subsection entitled "Position-dependent effects of layer mutation on SNARE assembly", in which the authors discuss single-molecule mechanics of 15 SNARE complex mutants, was very difficult to follow. We propose the following changes:

1) In Figure 4, show explicitly the name of the domain (NTD, MTD and CTD) to which each mutant belongs.

2) In the same figure, the current way in which FECs are marked is difficult to recognize. The colors and patterns are too similar. One way is to depict the corresponding transitions right next to individual FEC regions.

3) How is "MD+NTD" different from "MD_NTD"? The reviewers could not find proper explanations neither in the main text nor figure legends.

4) In Figure 5, each mutant trace is labeled with jargon, such as, "M46A CTD+MD_NTD". This labeling is nearly impossible to understand. It may be enough to depict the corresponding domains above the subpanel of Figure 5 (i.e., LC, CTD, MD and NTD). And the readers can then interpret the data as they are.

b) Figure 11 is also hard to follow. The authors introduced new States 10 and 11. Observation of the State 11 was attributed to folding of the C-terminal end of the t-SNARE proteins (called Tc in the manuscript) induced by Munc18-1 binding. An immediate question is whether such Tc structuring was observed in the presence of NRD. Also, the authors argued that the two partially-zipped states, State 9 and State 10, are different from each other (subsection “Munc18-1 induces t-SNARE folding”), which the reviewers could not understand. Because the NRD-truncated SNARE complex has less biological relevance, the authors may want consider moving Figure 11 to supplementary data. The data in Figure 9, Figure 10, Figure 12 and Figure 13 collectively make a smooth flow, but the data in Figure 11 seem at odds with those in the other figures.

---

## [Author Response]

Essential revisions:

The paper is, in some respects, two separate reports. In one, the authors characterize 15 different SNARE-complex mutants, previously studied by different methods and in different experimental setting. As the authors correctly point out, a coherent treatment by a uniform method is needed. In the other, the authors examine the effects of Munc18 on assembly and disassembly of single SNARE complexes. There are many new observations and insights, but the conclusions are less convincing than those of the first part (see below). Major comments: Overall: (a) The paper’s structure needs to be reorganized. Some important points are treated summarily (see last paragraph under "part 1", below), while some irrelevant ones are emphasized.

We agree with the comments made by the reviewers. Following the reviewers’ suggestion, we have made a major revision of our manuscript.

*(b) Please take care not to base structural interpretations on single distance measurements. While the models for states 1-8 are undeniably useful, and while the mutagenesis results as well as previous studies provide some additional support, the models should be presented and discussed more cautiously. Even the presence or absence of, for example, SNAP-25 in each of these states does not strike the reviewers as a foregone conclusion.*

We understand the reviewers’ concern here. The wide applications of optical tweezers in protein folding studies and our research experience in this field have led our belief that optical tweezers can be used to reliably determine the conformations and energies of protein folding intermediates. We have written a separate manuscript dedicated to our methods of data analysis, which was favorably reviewed in another journal. Basically, we combine our high-resolution single-molecule measurements, the crystal structure of the fully assembled SNARE complex, including its linear topology, and results of SNARE mutations, truncations, and protein binding to model the conformations of SNARE intermediates. We have described salient features of our methods, including key mathematical formulae, in the Data Analysis part of the Materials and methods section. In addition, we have added Figure 2—figure supplement 3 to describe how the extension and energy are calculated in our model. Furthermore, we have made three major changes to address the reviewers’ concern:

First, we have moved our description on SNAP-25 binding earlier in the text (subsection “Four stages of SNARE assembly in four domains”).

Second, we have elaborated our method of data analysis in the subsection “Energetics and kinetics of SNARE assembly”.

Finally, we have added a new section “Vc peptide induces folding of t-SNARE C-terminus and attenuates SNARE zippering” to describe effects of Vc peptide on SNARE assembly. As a result, we have added a new main figure, Figure 3. Results from the Vc peptide binding experiment confirmed our structural models shown in Figure 1.

In conclusion, these revisions, our previous results (Gao et al., 2012; Zorman et al., 2014), and results obtained from complementary approaches obtained in other groups (e.g., Li et al., 2014; Diao et al., 2012), corroborate the data analysis methods and the structural models derived by us.

*(c) As a specific example of interpretive ambiguity, the authors assume for the unfolding state 4 in Figure 1 that syntaxin (and probably also, SNAP-25) is partially disassembled up to layer +5. It is also possible, however, that the t-SNARE α-helices remain intact, while unfolding of VAMP2 progresses beyond the point shown, for example, up to layer -1 or -2. How do the authors differentiate between these possibilities? Since the C-terminal fragment of VAMP2 (VAMP2 49-96) induces tight folding of the C-terminal part of the t-SNARE proteins (many data in the previous literature and Figure 5 in this manuscript), addition of the C-terminal VAMP2 fragment should decrease the extension value of states 4 and 5 if the t-SNARE complex is indeed partially melted as the authors suggest. The authors might also (in part 2) repeat this experiment with Munc18, because they claim that Munc18, like the C-terminal VAMP2 fragment, chaperones the C-terminal end of the t-SNARE complex.*

The reviewers pointed out an important feature in our data analysis. In our model, SNAREs fold and unfold in a sequential and stepwise manner, between the well-defined fully assembled state (state 1) and the fully unfolded state (state 6). The total extension change is fixed by the extension difference between the two states. We essentially performed an interpolation to locate the conformations of intermediate states based on their measured relative extensions. Thus, an increase in the extension change of one folding step must be compensated by a decrease in the extension change of another folding step. In our analysis, we first determine the t-SNARE conformation (state 5) based on its extension relative to the completely unfolded state (state 6) and the Vc-peptide-bound state (state 8 in Figure 3). The t-SNARE transitions into both states are modeled as helix-coil transitions and the measured extension changes determine the lengths of the syntaxin helix involved in the transitions (please see Fig. S10 in Gao et al., 2012. This analysis reveals a frayed t-SNARE conformation as shown in Figure 1. This conformation is also supported by our extensive unpublished data on directly pulling the t-SNARE complex. As is hypothesized by the reviewer, we did observe extension shortening due to binding of the Vc peptide 49-96 (Figure 3) and Munc18-1 (Figure 11). Then we *simultaneously* fit the extensions of the intermediates 2-4 to the experimental measurements to determine the borders of these states. The fitting avoids the scenario proposed by the reviewer, because shifting the border between states 4 and 5 towards the N-terminus will shorten the extension change of NTD transition.

To address the reviewers’ comments, we have made the following changes:

1. Described our method of data analysis in the Materials and methods;

2. Added a new section, **“**Vc peptide induces folding of t-SNARE C-terminus and attenuates SNARE zippering”, to describe effects of Vc peptide on SNARE assembly. As a result, we have added a new main figure, Figure 3. Results from the Vc peptide binding experiment confirmed our structural models shown in Figure 1.

3. Added Figure 11 to specifically describe NRD-independent binding of Munc18-1 to the t-SNARE C-terminus (Tc).

(d) The suggested switching of Munc-18 interactions likewise appears to be arbitrary. The authors make no effort to justify that interpretation in terms of what is known about Munc18 structure and biochemistry. There are many ways in which two apparently distinct activities can be interpreted, without requiring distinct modes of interaction. The authors overstate the case by saying (rather pompously) "we discovered (sic) that Munc18-1 interacts with the SNARE bundle in at least two distinct modes".

We thank reviewers for the insightful suggestion. Following the suggestion, we have tried to integrate our observations to previous findings on Munc18-1 and also compared the differences in the Discussion part. As a result, we have made major revision in the Discussion part and also the model figure (Figure 14). In particular, we found that our independent observations on Munc18-1-mediated SNARE assembly and half-zippered SNARE complex are in excellent agreement with findings recently reported by Hughson and coworkers (Baker et al., 2015). Finally, we have revised the sentence mentioned by reviewers as follows:

“In conclusion, we observed that Munc18-1 interacts with the SNARE bundle in at least two modes and differentially regulates SNARE zippering: In an NRD-dependent mode, Munc18-1 stabilizes the half-zippered state, while in an NRD-independent mode, Munc18-1 structures Tc.”

*(e) The authors claim that they have "quantitatively verified the tight thermodynamic coupling between SNARE zippering and membrane fusion and revealed distinct functions of different SNARE assembly stages." They have done nothing of the kind. They have done a fine job of detecting intermediates in a useful and valuable model, in which individual SNARE complexes are pulled and relaxed, and they can (and should) be pleased with that accomplishment. But when several SNARE complexes are pulling on the same two membranes and the membranes are pulling back on them, as is the case for exocytic fusion (indeed, membrane fusion generally), then the relative heights of barriers, etc., might well be different. That difference doesn't diminish the value of the numbers provided here, but it does rule out egregious overstatement. Even the interpretation in the third paragraph, subsection “Energetics and kinetics of SNARE assembly underlying exocytosis”, seems stretched – why do the phenotypes of those mutations reveal "an essential role of independent MD assembly"? Yes, messing up the MD messes up the mouse, but how does even the correspondence to the kinetic observations reported here demonstrate "independent" (rather than "concerted", with the other segments) assembly?*

We are sorry for our overstatements. Fully accepting the reviewers' criticisms, we have revised the relevant sentences as “Combining our findings with previous results on exocytosis, we confirmed distinct functions of different SNARE assembly stages in synaptic exocytosis.”

“SNAREs couple their exergonic folding to membrane fusion (Sudhof and Rothman, 2009; Rothman, 2014). Such a thermodynamic coupling mechanism predicts that any perturbation in SNARE assembly impacts membrane fusion. Significant progresses have been made to test the prediction in the past two decades (Chen et al., 1999; Walter et al., 2010; Mohrmann et al., 2010). However, a quantitative test requires accurate and comprehensive measurements of both the energies and kinetics of SNARE assembly and the rates of membrane fusion for wild-type and various mutant SNARE complexes.”

and

“Other MD mutations L60A (+1) and L63A (+2) abolished the independent MD transition and thus changed the SNARE folding pathway. Yu et al. recently examined the impact of both mutations in synaptic exocytosis using cultured mouse neurons (Yu et al., 2015). They found that the two mutations abolished both spontaneous and evoked exocytosis, implying an important role of independent MD assembly in exocytosis. The comparison suggests that an obligate pathway of SNARE assembly may be required for membrane fusion.”

In addition, we have added many phrases to emphasize numerous previous reports on electrophysiological measurements of exocytosis. For example, in the Abstract:

“We found that SNARE layer mutations differentially affect SNARE assembly. *Comparison of their effects on SNARE assembly and on exocytosis* reveals that NTD and CTD are responsible for vesicle docking and fusion, respectively, whereas MD regulates SNARE assembly and fusion.”

Finally, we recognize that multiple SNARE complexes are required for membrane fusion, as is shown in our model figure (Figure 14) and had tried to integrate our single-molecule measurement to membrane fusion in our previous work (Zorman et al., 2014).

*Part 1: Although the authors basically performed the same experiments as in Gao et al. (2012), there seem to be two differences. The first is that unfolding in the C-terminal half of the SNARE complex, previously treated as one process, has now been resolved into CTD and MD unfolding (states 3 and 4), presumably because they did less temporal averaging of the data and plotted the extension traces over 1kHz. The second difference is that they moved the crosslinking site to layer -6. This difference likely separated NTD unfolding (state 5) from dissociation of SNAP-25 (state 6) more clearly than in Gao et al. (2012), in which the crosslinker was at layer -7. Thus, before dissociation of SNAP-25 at a higher force, they could see reversible NTD unfolding and refolding transitions. In Figure 2 and Figure 3, the authors used the high temporal (over 1 kHz) and spatial resolution provided by the optical tweezers to resolve detailed structural transitions occurring in a single neuronal SNARE complex – a substantial technical accomplishment. They resolve the C-terminal unfolding into two intermediates (3 and 4). The transition between the two involves unfolding (or refolding) of the middle part between the +2 and the zeroth layers (but this analysis may need to be refined). The mutations in the +1 or +2 layer accordingly affect the stability of these intermediates. Thus, the probability of observing intermediate 3 is selectively reduced (i.e., the unfolding process pauses less frequently at the +2 layer).*

We thank the reviewers for correctly summarizing our main findings presented in the manuscript.

*The authors claim that "the middle domain plays a pivotal role in robust and correct stage-wise SNARE assembly, and its mutation leads to changes in SNARE assembly pathway and accuracy" and that "the MD serves as a checkpoint to further ensure proper SNARE assembly as a master switch to control CTD assembly". It is hard to understand how the reduced observation of the intermediate 3 (observed for the L60A and L63A mutants) supports these rather strong arguments. In the third paragraph of the subsection “Energetics and kinetics of SNARE assembly underlying exocytosis”, the authors allude to unpublished results, but this citation is not an appropriate way to justify the arguments. The authors should either weaken these claims or provide experimental evidence to support their arguments.*

Following the reviewers’ suggestion, we have weakened our claim on the significance of the MD, as is described in our preceding replies. During revision of this manuscript, the unpublished work referred in our previous manuscript has been published (Yu et al., 2015). We have updated the citation. The major and common change caused by L60A and L63A is a combination of both CTD and MD into one transition unit, which demonstrates a change in the SNARE assembly pathway. Therefore, we suggest that a conserved SNARE folding pathway, including an independent MD folding, may be required for membrane fusion.

*Part 2: Data showing different effects of Munc18 are mixed confusingly together, making part 2 extremely hard to follow, even for a reader who knows the field. In particular, in Figure 5 and Figure 6, rearrangements of the figure subpanels would improve readability. For example, the subpanels, Figure 5 and Figure 6 report that Munc18 helps reassembly of the SNARE complex after complete unfolding of the t-SNARE pre-complexes, so that these panels can be grouped together to make one figure. On the other hand, the subpanels including Figure 5 to G and Figure 6 to D report stabilization of the half-zipped SNARE complex by Munc18, which may be grouped together to make another figure.*

We thank the reviewers for pointing out our poor figure organization. Following the reviewers’ suggestion, we have made major efforts to re-organize the figures in the whole revised manuscript. We make sure that each figure focuses on one subject matter. We believe these changes substantially improve the readability of the second part of the manuscript.

*From the extension difference between states 8 and 5 (*Figure 6*), observed only in the absence of NRD (N-terminal Regulatory Domains), the authors argue that Munc18 would bind to the C-terminal part of the t-SNARE proteins and induce a coil-to-helix transition. But then, state 4 with bound Munc18 should have a smaller extension than state 4 without Munc18.*

The reviewers would be right if VAMP2 were free to zipper in this case. However, the average extension change upon NTD zippering on Munc18-1 bound t-SNARE complex (between states 10 and 11 in Figure 11) is smaller than the corresponding extension change in the absence of Munc18-1 (between states 4 and 5 in Figure 2). This indicates that VAMP2 can only zipper to a position N-terminal to -1 layer in the presence of bound Munc18-1, which prevents VAMP2 from zippering to a position corresponding to state 4. We designated this new position as state 10 in Figure 11 and illustrated its different position in Figure 11. Note that due to its short lifetime, state 4 is not always clearly seen in some extension-time trajectories exhibited at 1000 Hz, as is stated in the main text.

Figure 7 and the accompanying discussion are not very clear, and the reviewers found the conclusions unconvincing. For example, the opening and closing of syntaxin induced by Munc 18 (Figure 7) is intriguing, but then why do authors not observe reversible transitions between open and closed syntaxin in the more direct pulling study shown in Figure 7?

We appreciate that the reviewers pointed out the difference. Closed inspection showed that the transition in this new pulling direction is indeed reversible, as is seen in one of the force-extension curves shown (Figure 13, FEC #2). We have now added a new FEC to better exhibit the reversible transition (FEC #4). The transition rate is much lower than that seen in the previous construct (Figure 12). We have clarified the difference in the main text as follows:

“The transition was reversible (FECs #4 and #2), but with much smaller transition rate than that seen in the previous construct (Figure 12). As a result, the extension flicking during pulling or relaxation was not seen in some FECs, such FEC #3 in Figure 13.”

“The different transition kinetics seen in the two syntaxin constructs is not surprising, because the syntaxin molecule was pulled in two approximately perpendicular directions (compare Figure 13 to Figure 12). It is known that protein folding kinetics, but not folding energy, strongly depends on the pulling direction (Gao et al., 2011).”

*Overall recommendation: 1) Present the structural models for the kinetic states more cautiously and explain any alternatives.*

Yes, we have done so. Please see our response to the Major comments (b) and (c).

*2) Support the interpretation of the role of the MD with data (and pay attention to other comments under "part 1", above).*

Yes. Please see our reply to the last comment in Part 1.

*3) When revising what we have called "part 2", comply with each of the three paragraphs under "part 2", above.*

Yes. Please see our replies to the comments in Part 2 above.

*4) Provide details (in the Methods) for the nonlinear least-square fitting that converts dwell times to deltaG (or deltaG double dagger).*

Yes, we have provided details in Methods for the nonlinear least-square fitting to derive the energies of intermediate states and transition states.

[Editors' note: further revisions were requested prior to acceptance, as described below.]

Essential revisions:

*A) The subsection entitled "Position-dependent effects of layer mutation on SNARE assembly", in which the authors discuss single-molecule mechanics of 15 SNARE complex mutants, was very difficult to follow. We propose the following changes: 1) In Figure 4, show explicitly the name of the domain (NTD, MTD and CTD) to which each mutant belongs.*

Following the reviewers’ suggestion, we have explicitly shown the name of the domain to which each mutant belongs. In addition, we have color-coded the mutation names to blue, green, and red based on their associated NTD, MD, and CTD, respectively. The same color-coding is applied to mutation names in Figure 5 and Figure 6 and Figure 5—figure supplement 1, Figure 5—figure supplement 2.

2) In the same figure, the current way in which FECs are marked is difficult to recognize. The colors and patterns are too similar. One way is to depict the corresponding transitions right next to individual FEC regions.

We have improved the labeling by adding domain names next to the corresponding transitions in FECs, as the reviewers suggested. In addition, we have simplified the labeling by only showing the abnormal transitions of the mutant SNARE complexes compared to the wild-type.

3) How is "MD+NTD" different from "MD_NTD"? The reviewers could not find proper explanations neither in the main text nor figure legends.

The “MD+NTD” transition means sequential MD and NTD transitions in an overlapping force range, whereas the “MD_NTD” transition indicates two-state transition of MD and NTD as a combined new domain. To address the reviewers’ concern, we have added Figure 4—figure supplement 1 to illustrate the change in SNARE folding pathway. In addition, we have clarified our nomenclature in several places in the main text, as well as in the legend to Figure 4.

4) In Figure 5, each mutant trace is labeled with jargon, such as, "M46A CTD+MD_NTD". This labeling is nearly impossible to understand. It may be enough to depict the corresponding domains above the subpanel of Figure 5 (i.e., LC, CTD, MD and NTD). And the readers can then interpret the data as they are.

As is described in our reply to the preceding question, we have depicted the difference between sequential transitions and cooperative transitions in Figure 4—figure supplement 1, using “MD+NTD” and “MD_NTD” as an example. Combining with our clarified definition of the nomenclature in the main text, we think the nomenclature can be better understood. These nomenclatures are well consistent with our labeling of different state positions in the extension-time trajectories (Figure 5).

b) Figure 11 is also hard to follow. The authors introduced new States 10 and 11. Observation of the State 11 was attributed to folding of the C-terminal end of the t-SNARE proteins (called Tc in the manuscript) induced by Munc18-1 binding. An immediate question is whether such Tc structuring was observed in the presence of NRD. Also, the authors argued that the two partially-zipped states, State 9 and State 10, are different from each other (subsection “Munc18-1 induces t-SNARE folding”), which the reviewers could not understand. Because the NRD-truncated SNARE complex has less biological relevance, the authors may want consider moving Figure 11 to supplementary data. The data in Figure 9, Figure 10, Figure 12 and Figure 13 collectively make a smooth flow, but the data in Figure 11 seem at odds with those in the other figures.

We thank the reviewers for pointing out our confusing description regarding Figure 11. We have made significant revision in this part to address the reviewers’ comments. First, the Tc structuring was indeed observed in the presence of NRD, although with much lower probability than in the absence of NRD. We have added Figure 11—figure supplement 3 to show Munc18-1-induced folding of Tc in the SNARE complex containing NRD. Second, we agree with the reviewers that our original description on the difference between state 9 and state 10 is confusing. We have revised the relevant sentences. Basically, the two states differ in their Munc18-1 binding sites on the SNARE complex, the VAMP2 zippering states, and mechanical stabilities. Third, we have added a few sentences at the beginning of the section to rationalize the experiments described in this section and emphasize their biological relevance. Previous experiments show that, although the NRD is initially required to recruit Munc18-1 to bind the trans-SNARE complex, the NRD becomes dispensable for the subsequent membrane fusion, indicating the existence of an NRD-independent Munc18-1-SNARE binding mode. The NRD-independent, Munc18-1-mediated Tc folding discovered by us is consistent with the predicted binding mode. We believe this binding mode is important, because it shares the same mechanism by which Vc and Vn peptides enhances SNARE-mediated membrane fusion in vitro. Based on these thoughts and our clarifications, we kept Figure 11 as a main figure. In addition, this figure strengthens our argument for multiple Munc18-1 binding modes and their roles in SNARE assembly and membrane fusion depicted in Figure 14.